# Measuring the popularity of football players with Google Trends

**Pilar Malagón-Selma**[1]*, **Ana Debón**[1], **Josep Domenech**[2]

**1** Centro de Gestión de la Calidad y del Cambio, Universitat Politècnica de València, Valencia, Spain,
**2** Departamento de Economía y Ciencias Sociales, Universitat Politècnica de València, Valencia, Spain

* pimasel@doctor.upv.es

**Data Availability Statement:** Our primary data source, as well as all code associated with this project is available at https://github.com/Malagon-Selma/Market-Value-BIG5-2018-2019. In particular, the data was collected through the

## Abstract

Google Trends is a valuable tool for measuring popularity since it collects a large amount of information related to Google searches. However, Google Trends has been underused by sports analysts. This research proposes a novel method to calculate several popularity indicators for predicting players' market value. Google Trends was used to calculate six popularity indicators by requesting information about two football players simultaneously and creating popularity layers to compare players of unequal popularity. In addition, as the main idea is to obtain the popularity indicators of all players on the same scale, a cumulative conversion factor was used to rescale these indicators. The results show that the proposed popularity indicators are essential to predicting a player's market value. In addition, using the proposed popularity indicators decreases the transfer fee prediction error for three different models that are fitted to the data using the multiple linear regression, random forest, and gradient boosting machine methods. The popularity indicator *Min*, which is a robust reflection of the popularity that represents a player's popularity during the periods when they are less popular, is the most important popularity indicator, with a significant effect on the market value. This research provides practical guidance for developing and incorporating the proposed indicators, which could be applied in sports analytics and in any study in which popularity is relevant.

## Introduction

With at least 158 years of history, football continues to be the hegemonic sport of today's society. The sports industry is a place where most interests, from political interests to public interests, converge, and traditional media and social media have helped to increase football's popularity worldwide [1, 2]. Indeed, the popularity of this sport has caused football clubs to generate increasing amounts of revenue. According to the consulting company Deloitte, the 20 clubs with the highest turnover (all of which belong to the 'Big Five' European football leagues) exceeded €9.200 million in revenue in the 2021/22 season, marginally below pre-COVID revenues of €9.283 million in 2018/19 [3].

Football players have already been recognised as accounting assets for clubs [4], and their valuation is an essential indicator of the financial value of teams [5]. Thus, from both sports and business points of view, the revenues of football teams are often closely related to the

Internet websites of Google Trends (trends.google. es), WhoScored (whoscored.com), FBref (fbref. com), and Transfermarkt (transfermarkt.es).

**Funding:** P.M-S. Economic support through the FPI-UPV scholarship (PAID-01-19) to the Universitat Politècnica de València. http://www.upv. es/entidades/VINV/indexc.html P.M-S, A.D., and J. D. Grant PID2019-107765RB-I00, funded by MCIN/AEI/10.13039/501100011033. The funders had no role in study design, data collection and analysis, decision to publish, or preparation of the manuscript.

**Competing interests:** The authors have declared that no competing interests exist.

teams' investment in their main assets: the players. In 2009, Real Madrid C.F. made the most expensive transfer up to that date, paying €94 million for Cristiano Ronaldo. However, more than ten years later, this record was broken repeatedly, and the world's three most expensive transfers are those of Ousman Dembélé (€140 million), Kylian Mbappé (€145 million plus €35 million commission), and Neymar Jr (€222 million). Given the impact that a signing of this magnitude can have on a football club, media professionals, managers, analysts, and other experts have tried to determine the factors that contribute most to the estimated market value of a player. According to Franceschi et al. [5], researchers have recognised the empirical proximity between players' transfer fees and their market value. Therefore, these values are comparable [6] since they store similar information and are influenced by the same variables. The transfer fee is 'the actual price paid on the market' [7] by a football team for a player at a given time, and it is rarely available to the public [5]. Thus, to solve the problem of the lack of information on the actual transfer fees, researchers have started paying attention to websites that offer estimates of the market value of players [5, 7, 17]. An example is Transfermarkt, which, although the website itself explains that its goal is not to predict player transfer fees but to provide the 'expected value of a player in a free market' [8], has gained great prestige not only among football industry professionals (coaches and journalists) but also among scientists who have found this website helpful for estimating the market value [7, 9, 10].

In this context, researchers from different areas of knowledge have begun to specialise in valuating players and studying the factors that affect the market value to predict transfer fees [7, 9, 11]. The player's performance, position (forward, midfielder, defender, or goalkeeper), club, and physical characteristics (e.g. height and age) are the variables most often used in such studies [7, 12, 13].

Moreover, with the emergence of social networks such as Instagram and Twitter, the brand images of football players have benefited. A clear example is Cristiano Ronaldo, who in 2021 had 308 million followers on Instagram (making him the person with the most followers in the world), charged $1.6 million per sponsored post, and earns more than 40 million dollars per year from publications alone [14]. Now, two years later, his followers exceed 590 million and he is still the person with the most followers in the world. Thus, it seems reasonable to consider popularity as an essential factor that affects the transfer rate, especially since this information is open and easy to find.

In particular, the (probable) effect of popularity on the market value, which can be used to predict transfer fees, has not gone unnoticed. In recent years, researchers have incorporated popularity measures such as the number of followers on social media [7, 15], the number of Google hits (defined as the number of links reported by Google when the player's name is searched) [13, 15], and Wikipedia views [7, 10] to estimate the market value and predict the transfer fee of a football player, showing that popularity has a statistically significant effect. Müller et al. [7] also used other measures of popularity, such as the number of times a player's name appeared in Reddit posts, the number of YouTube videos in which a player appears, and information provided by Google Trends (GT).

In this context, GT is a very interesting tool that allows us to measure the interest in a topic or person over time [16]. Specifically, GT provides a time series that allows the granular measurement of the number of Google searches; Müller et al. [7] used the GT time series average (GTA), which was obtained for each player individually, as a popularity indicator. However, although this variable could be closely related to popularity, the GTA was the only popularity-related variable that was not statistically significant. The non-statistical significance of the GTA variable could be explained by the way it is calculated. GT provides topic-dependent normalised indexes from 0 to 100 rather than a time series of absolute searches. The values

provided by GT are unsuitable for a direct comparison between different players since these values were obtained individually for each player.

Therefore, there is a gap in the existing research concerning how to use GT to quantify the influence of popularity on the market value. First, some of the measures described above contain information about a specific moment, and others, such as GT information, need to be used and summarised appropriately. Second, some of the variables (probably) store redundant information; for example, the number of visits to a player's Wikipedia page is closely related to the number of Google searches, as most of the time, Google is used to access Wikipedia. Third, although researchers have found that popularity has a statistically significant influence on the market value, compared to other variables, it is not as widely studied [5]. Fourth, authors who have analysed the effect of popularity have not studied how much of the transfer fee is explained by it.

The main contribution of this paper is to propose a novel way of calculating popularity indicators from GT time series. Since GT is a granular measure of the number of Google searches, these indicators allow us to summarise and collect information on other popularity variables related to Google (e.g. the number of Wikipedia views, YouTube videos, or Google hits). Thus, using GT simplifies the data collection process, and the popularity indicators do not contain redundant information. In addition, the use of GT allows us to avoid the problem of missing values. To demonstrate the usefulness of the proposed popularity indicators, they will be applied to train models to predict the estimated market value of players not signed during the summer market of the 2018–2019 season. The model error will be assessed using the transfer fees of players signed during the summer market after that season. In addition, the effect of popularity on the transfer fee will be measured by quantifying the decrease in the prediction error.

## Background

### Estimation of market value

One of the challenges of today's football industry is the valuation of the players. In this context, websites that fans can use to estimate the market value of players have been developed. For example, the Transfermarkt platform is a website that offers information on the transfer market and allows its users to rate professional footballers. However, the valuation of the players is not carried out from a democratic perspective but is governed by the 'principle of the judge' [13]. According to Herm et al. [13], a community member can propose the value of a player as long as the objective data support this value; for example, it could be based on the player's performance variables or similarity with another player. After the initial proposal, the members begin a discussion about the proposed value. However, Transfermarkt does not make the final decision democratically. A group of 'judges' (community members with greater weight than the rest) estimates the market value from the users' proposals and knowledge. Although this form of assessment could be shown to be ineffective or subjective, according to Herm et al. [13], the task of the 'judges' is more complex than it seems since they have to filter, weigh, and add information to the justifications given by the users to adjust the market value. In addition, the work of the 'judges' is constantly reviewed, as each time a player is transferred, the estimated value is compared to the actual transfer fee.

Numerous studies have acknowledged Transfermarkt as a useful approximation to players' market value. However, such a crowdsourcing approach has its limitations. In this respect, Müller et al. [7] point out that the objectivity of the community members, on whose opinions the valuation is based, cannot be guaranteed. The process of aggregating user valuations lacks homogeneity, making it difficult to reproduce the results. Moreover, the estimation of market

value tends to have higher variability for less popular players receiving less attention and valuations. More recently, Franceschi et al. [5] stress the subjectivity already mentioned by [7] and emphasize the conceptual difference between the transfer fee and the estimated market value provided by platforms like Transfermarkt. Nevertheless, researchers have recognized the empirical proximity between these measures [5, 7, 13, 17, 24, 27].

## Explanatory variables for estimating market value

This subsection discusses the most common variables used in the player valuation process and their significance. Researchers have identified four principal groups of variables for analysing a player's market value: the player performance, player characteristics, labour and club characteristics of the players, and player popularity.

**Player characteristics.** Previous research has considered player characteristics, e.g. a player's *age*, *height*, *footedness*, *nationality*, and *position*; these quantities represent physical or market information, as they are variables related to the market value [7, 9, 11, 18, 19].

The variable *age* ($age^2$ is also used) is a variable that most studies include and generally find statistically significant [7, 9, 13]. Müller et al. [7], who used $age^2$, found that this variable was statistically significant, with a negative effect on the players' market value. Herm et al. [13] found the same effect, although they used an age-corrected linear function. Behravan and Razavi [9], who considered how independent variables affect the player valuation for different *positions*, found that *age* influenced the market value for all positions except goalkeepers. In addition, some studies have even used the quarter of the year in which each player was born as an explanatory variable and found evidence that it is related to the market value [20, 21]. According to Felipe et al. [20], the most valued players are attacking midfielders born during the first quarter. Gyimesi and Kehl [21] studied the effect of the relative *age* on the market value, concluding that it is higher at younger ages and decreases as the player ages. However, some research found a non-statistically significant relationship between *age* [12, 22] or $age^2$ [12] and the market value. Finally, Franceschi et al. [5] considered that the importance of the *age* variable and its positive impact (negative when the quadratic form is used) on the market value of players is due to the relationship between the experience and *age* of the players, i.e. the older the player, the more experience they have, but the older the player, the lower the player's physical potential.

Most researchers agree that the *position* is an influential variable for the players' market value [6, 7, 10, 12]. Garcia-del Barrio and Pujol [12] used the *position* to control for heterogeneity associated with a player's position on the field. Their results showed that forwards have a higher valuation than goalkeepers (the reference used in their regression models). Frick [6] determined that the *position* significantly influences player salaries, with goalkeepers being the players with the lowest salaries. Müller et al. [7] considered four multilevel models, which included the *position* as a random effect, to capture the dependence between the market values of the players who played in the same position. Other authors, such as Behravan and Razavi [9] and Majewski [22], estimated the market value of players by grouping them according to their positions and predicting the market value for each group of players independently. In addition, the explanatory variables used to predict the players' market value were selected depending on their position on the field (goalkeeper, defender, midfielder, or forward) [9, 22].

Furthermore, Herm et al. [13] and Bryson et al. [23] argued that the ability to 'two-foot' could improve a player's market value, as it allows the player to play in different positions. Behravan and Razavi [9] also considered it necessary for forwards to be able to use both feet; in particular, they reflected on the importance of being able to use the 'weak foot' accurately. However, Garcia-del Barrio and Pujol [12] and Müller et al. [7] did not find that the footedness variable was statistically significant.

Finally, previous analyses have examined the effect of *nationality* on the market value [7, 12, 19]. Garcia-del Barrio and Pujol [12] found that European players tended to be overvalued in LaLiga (Spanish first division), while non-European players were undervalued. Pedace [19], who analysed discrimination based on *nationality* in the Premier League (English first division), found evidence that South American players were overvalued. Müller et al. [7] also included the *continent* of origin as a random effect to capture the dependence between the market values of players born on the same continent.

**Player performance.**   The player performance variables measure players' actions during the season. The *playing time* is a standard statistically significant variable researchers use to determine the market value [7, 10, 18]. In addition, studies have found a positive and significant relationship between the number of *goals* and *assists* and the market value [7, 10, 13, 22]. The consistency and importance of both variables have led some researchers to develop new study methods. For example, Herm et al. [13] developed the scoring variable, which is calculated as the number of *goals* and *assists* corrected by the average for the player's position. According to their results, the higher the number of goals or assists scored, the higher the market value. Majewski [22] developed a synthetic variable using the sum of the goals and assists, obtaining a positive and statistically significant result.

Behravan and Razavi [9] developed the *shooting skill* variable, which stored information on accuracy, instead of studying the relationship between the number of goals and the market value. The results indicated that the shooting skill was essential to the market value for defenders and forwards.

Regarding the defensive variables, the *number of interceptions*, *yellow* and *red cards*, and *fouls* have also been occasionally used [7, 22]. Majewski [22], who introduced *yellow* and *red cards* into the model, found that neither variable was statistically significant. According to Müller et al. [7], only the number of *yellow cards* was statistically significant. According to Franceschi et al. [5], although the number of *yellow cards* and the number of *red cards* are easy and inexpensive variables to calculate, their use may not be recommended due to their repeated lack of significance.

Even though these variables are used more infrequently, previous research has considered the *passing accuracy* [7, 9, 13], the *number of dribbles* [7, 9], and the *number of duels* [7, 13]. Herm et al. [13] defined the variable accuracy as the percentage of successful passes corrected for the average for the player's position. Based on their results, the more accurate the player's passes are, the higher their market value will be. Similarly, Herm et al. [13] calculated the number of won duels, or the save-to-shots ratio for the goalkeeper, corrected for the average for each position, and found that it had a significant effect on the players' market value. Specifically, Müller et al. [7] considered the following explanatory variables of the market value: the *number of passes*, *successful passes*, *number of aerial duels*, *successful aerial duels*, *number of dribbles*, and *successful dribbles*. However, only the *number of passes*, *successful passes*, the *number of aerial duels*, and the *number of dribbles* were positive and statistically significant. Behravan and Razavi [9] studied the effect of the *passing* and *dribbling* abilities of the players according to their positions. They found that dribbling should be considered when the relationship of the variables that affect the market value of forwards is studied. Furthermore, their results showed that *long passes* (in the case of defenders and midfielders) and the *passing* ability (in the case of forwards) were essential factors in estimating the players' market value.

**Labour and club characteristics of the players.**   The group of variables related to the labour and club characteristics of the players includes variables related to a player's employment contract (e.g. the *length of the contract*), the league in which the player competes, previous player valuations, and information that is not directly related to the player's performance but is related to their club's performance (e.g. the *FIFA ranking of the teams*).

Previous analyses have examined the effect of the *contract length* on players' market value and found evidence that the longer the contract expiration date, the higher the transfer fee [18, 24–27]. Furthermore, Franceschi et al. [5] argued that, given that at the end of a player's contract, their transfer value is zero, if the time remaining until the end of the contract is not included in market value prediction models, relevant information affecting the transfer fee of players could be lost.

Other valuations, such as the *previous market value*, have also been used to predict the market value [7, 10]. To estimate the player's market value after a given season, Müller et al. [7] used, as an explanatory variable, the log transformation of the Transfermarkt estimate of the player's market value from the end of the previous season. Similarly, Singh and Lamba [10] used the *previous market value* estimated by Transfermarkt to predict the current market value of players. In both investigations, the variables were positive and statistically significant.

The *league* in which the player competes is a categorical variable introduced as a random effect to capture the non-independence between players' market values and the league in which they compete [7, 27]. Felipe et al. [19], who analysed the interaction of different explanatory variables with the Transfermarkt market value, also took the *league* into account. According to their findings, there were differences between the market values of players competing in the Premier League and the market values of players in other leagues. Gyimesi and Kehl [21] obtained similar results and found statistically significant differences in the market values of similar players among the Big Five.

Furthermore, previous researchers considered the impact of players' *presence in European championships* and on their *national teams* in predicting their market value [10, 12]. Garcia-del Barrio and Pujol [12] counted how many matches a player had played in Europe (considering both the *Champions League* and the *UEFA Cup*) and how many matches the player had played for their national team. For both variables, quadratic terms were also considered. Garcia-del Barrio and Pujol [12] found that players who had competed in Europe and with their national teams had a significantly higher market value. Singh and Lamba [10] used the valuations of players provided by the FIFA 2018 game as an explanatory variable and found that this value impacted the players' market value.

Finally, how a *local* or *national team's reputation* influences players' market value has also been analysed [9, 22]. Majewski [22] considered the *FIFA national team ranking* as an integer from 1 (the best national team) to 209 (the worst national team). According to their results, since the coefficient of this variable was negative and statistically significant, a lower ranking of the national team decreased the value of the players. According to the study carried out by Behravan and Razavi [9], the *reputation of the national team*, measured based on the FIFA world ranking, is important for estimating defenders' and midfielders' market value.

**Player popularity.** While the characteristics and performance of players have helped predict their transfer fees, they are not perfect measures of a football player's market value; researchers such as Franceschi et al. [5] have highlighted that variables such as contracts, youth academies, and popularity have strong theoretical justifications and promising empirical results that indicate that they can better explain the transfer market. In the same way, previous researchers have pointed out that popularity-related data are an interesting complement that can improve a player's valuation [7, 12, 13, 15]. Garcia-del Barrio and Pujol [12], who incorporated popularity ratings into their study, expressed their concern that (before then) no study related to the sports industry had included popularity variables, as these turned out to have an important role in helping to explain economic behaviour in the sports industry. Herm et al. [13] emphasised that it is necessary to study variables related to popularity, as these variables impact the market value of athletes. Hofmann et al. [15] highlighted that managers are interested in knowing whether performance or popularity is more important in boosting the

players' market value. Müller et al. [7] concluded that not only player characteristics and performance but also measures of player popularity could help football teams make predictions concerning the evolution of the market value.

Previously, the popularity of players has been studied using different variables. *Hits on Google* and social media (*Facebook*, *Twitter*, and *Instagram*) are the two variables most commonly used to analyse the impact of popularity on the market value. Garcia-del Barrio and Pujol [12], Herm et al. [13], and Hofmann et al. [15] used information on the *number of hits on Google* when a player is searched for using their full name; this variable was statistically significant and had a positive coefficient. In addition, Hofmann et al. [15] considered the *number of followers on social media*. These studies showed the importance of both of these popularity measures in predicting a player's transfer fee. Müller et al. [7], in addition to social media variables, incorporated *Reddit posts*, *Wikipedia views*, *YouTube videos*, and *Google Trends*. According to Müller et al. [7], all social media variables were statistically significant except for *Google Trends*. Finally, Singh and Lamba [10] found that *Wikipedia views* enhanced the prediction accuracy of football players' transfer fees.

## Use of popularity as an explanatory variable

Today, the superstar theories put forward at the end of the 20th century are still valid in an industry that wonders whether the player makes the brand or the brand makes the player. According to Rosen [28], a personal brand arises from human talent, while Adler [29] argued that popularity is the main driver of a superstar. Under this premise, Müller et al. [7] stressed that popularity is what sells shirts and seats regardless of the player's performance on the field. Similarly, Hofmann et al. [15] highlighted the correlation between a player's brand image and the market value of the player. Hofmann et al. [15] argued that most clubs own the branding rights of their players, which brings additional revenue to the team. However, the concept of image rights (or branding rights), understood today as 'publicity rights' [30], is not subject to legal rules but depends on contractual negotiations between players, agents, and teams [31]. As Hofmann et al. [15] indicated, it is typical for a football club to own the publicity rights of the players, who, in most cases, are required to cede the use of trademarks or sponsorships for marketing and commercial activities that provide financial profitability to the team [32]. However, not all clubs exert the same pressure on the control of publicity rights; while the most powerful teams (e.g. Arsenal, Manchester United, and Real Madrid) demand complete control, others agree to obtain a minor portion of profits from the commercial exploitation of players' publicity rights [30]. Thus, while 'superstars' strive to maintain complete control over their image rights, average football players are often denied this privilege. Therefore, given the impact of a transfer on the economy of the clubs, it seems necessary to study the specific effect of popularity on the transfer fee. In this context, popularity has been measured in different ways, as mentioned above.

Several studies highlight the statistical significance of popularity regardless of the season, league, or player being analysed. Garcia-del Barrio and Pujol [12] used data from 369 football players who competed in LaLiga during 2001/2002. Instead of the classical performance variables, they used two variables that represented the productivity of the player measured by a performance index provided by Marca (a Spanish newspaper) and the Fantastic league. Additionally, they considered the position, the age, and two dummy variables for foreign players: non-European and non-Spanish. Regarding popularity variables, Garcia-del Barrio and Pujol [12] used the number of Google hits and created three dummy variables: the top 5, 10, and 20 most popular players (they chose the players according to the number of Google hits). Then, they created a baseline model, including the players' characteristics and performance and the

number of Google hits, and they created a second model that also included the popularity dummy variables.

Müller et al. [7] collected information about 4,217 players who competed in the Big Five leagues over six seasons. They applied a multilevel regression method to estimate the market value of the players. They used a database composed of variables related to player characteristics, performance, and popularity. Müller et al. [7] used different popularity measures; the GTA was the only popularity variable studied that was not statistically significant. Like Garcia-del Barrio and Pujol [12], Müller et al. [7] considered a baseline model and three additional models, which included player characteristics, performance, and popularity.

Singh and Lamba [10] used data about players who participated in EURO 2012 to analyse how their market value was affected by their performance in the tournament. They applied different machine learning algorithms to perform a predictive analysis using a database composed of performance, popularity, and market value variables. The research objective was to find the group of variables that contributed most to the prediction of the market value. As a popularity measure, Singh and Lamba [10] only considered the number of Wikipedia views.

Hofmann et al. [15] conducted two analyses to study the effect of performance attributes on players' brand images. The first was an empirical analysis that measured the effect of performance attributes (attractiveness, appearance, temperament, and media appearance) on the brand images of 25 players. The second analysed the impact of measures of player popularity (Google hits and followers on Instagram and Facebook) on the market value. For the second analysis, they used a database of 316 football players. Hofmann et al. [15] concluded that while brand and performance attributes are positively correlated with the market value, the player popularity measures they used to conduct the study only moderately explained the popularity.

Although researchers have found that popularity is essential to predicting players' transfer fees, most have not studied its specific impact in depth; this impact can be measured by quantifying the increase in the market value due to the player's popularity. Thus, throughout this paper, we will study and quantify the specific effect of popularity on the predicted transfer fee using GT information.

GT is a valuable tool that contains information on all popularity variables related to Google. In contrast to previous researchers, who collected data from different internet sources (e.g. the number of visits to Wikipedia, YouTube videos, or Google hits), using GT simplifies the data collection process since a single variable contains most of the information related to Google. In addition, using a single variable that stores all the information avoids the problem of multicollinearity. Another essential advantage of GT is that while most of the variables described above contain information about a specific point in time, GT contains information about a time frame. Finally, GT allows us to study the evolution of a player's popularity without having to deal with the problem of missing values.

Although GT is a helpful tool for measuring a player's popularity, this variable has been underused and misused. Therefore, this study will analyse the extent to which the market value is related to GT measures and explore whether the information provided on the website can be used to predict the market value of a player. Specifically, we will summarise the GT information in several indicators that are easy and inexpensive to calculate.

## Construction of popularity indicators

GT is a tool that measures the interest in a topic or person by providing a time series. This time series contains relative granular (daily, weekly, etc.) values of topic popularity, normalised from 0 to 100; the value 100 occurs in the period with the highest number of searches on the topic [16]. Normalisation makes it easier to study the evolution of the topic's popularity.

However, it does not allow comparisons between topics since the time series of each topic is normalised individually. According to Rogers [16], one way to put a topic's search interest in perspective could be to add additional topics. Thus, by considering two topics simultaneously (in our case, two players), the results of both series are jointly normalised with respect to the maximum of both topics in the period studied. Therefore, both GT series are on the same scale, and it is possible to compare them. Henceforth, the normalised GT time series obtained when two players are compared will be referred to as the GTN. Moreover, it should be noted that, as the main idea is to obtain the popularity indicators of all players on the same scale, in the end, these indicators have to be rescaled to the same player.

Unfortunately, the GTN is reported as integers instead of real numbers. Thus, if a famous player is compared to a very unpopular player, the GTN is 0 for the latter. Accordingly, the normalisation of the GTN makes it difficult to compare players of unequal popularity. To solve this problem, we propose using different reference players (RPs) according to their relative popularity. Moreover, since the notoriety of a player depends on the player's position, the players were also ranked by position.

Table 1 shows the reference players ($RP_{lj}$), where the row $l = 1, 2, 3$ denotes the layer popularity and the column $j = f, m, d$ denotes the position.

For forwards, midfielders, and defenders, the most popular player was chosen as $RP_{1j}$. Then, $RP_{2j}$ was chosen as the most unpopular player of those players whose GTA was one relative to $RP_{1j}$, and $RP_{3j}$ was chosen similarly by considering $RP_{2j}$. $GTN(player_{l,j}, RP_{lj})$ denotes the normalised GT time series obtained by comparing $player_{l,j}$ with a specific reference player $RP_{lj}$.

After choosing $RP_{1,j}$, it was necessary to rescale all players to $RP_{1,f}$. First, conversion factors $CF_{1,f}$ for each layer $l = 1, 2, 3$ and position $j = f, m, d$ were obtained as follows:

$$CF_{lj} = \begin{cases} \dfrac{max(GTN(RP_{lj}, RP_{1j}))}{100} & l = 1 \quad j = f, m, d, \\[2em] \dfrac{max(GTN(RP_{lj}, RP_{l-1,j}))}{100} & l = 2, 3 \quad j = f, m, d. \end{cases} \quad (1)$$

Second, the cumulative conversion factors $CCF_{lj}$ of these factors ($CF_{lj}$ from Eq 1) allow all the players to be rescaled to $RP_{1,f}$:

$$CCF_{lj} = \prod_{r=1}^{l} CF_{rj} \quad l = 1, 2, 3 \quad j = f, m, d. \quad (2)$$

Then, Eq 3 was defined to rescale the player popularity, taking into account the corresponding $CCF_{ij}$ from Eq 2:

$$GTN(player_{l,j}, RP_{1j}) = GTN(player_{l,j}, RP_{lj}) \times CCF_{l,j} \quad l = 1, 2, 3 \quad j = f, m, d. \quad (3)$$

Table 2 shows the *CF* and *CCF* values calculated for each layer and position.

**Table 1. Reference players according to their popularity layers and positions.**

| Popularity layer | Forward | Midfielder | Defender |
|---|---|---|---|
| 1 | Cristiano Ronaldo | Paul Pogba | Sergio Ramos |
| 2 | Franck Ribéry | Fabián Ruiz | Kamil Glik |
| 3 | Kevin Lasagna | Yannick Gerhardt | Diego Rico |

**Table 2. Conversion factors (*CF*) and cumulative conversion factors (*CCF*) for the players according to their popularity layers and positions.**

| Popularity layer | Factor | Forward | Midfielder | Defender |
|---|---|---|---|---|
| 1 | CF | 1 | 0.15 | 0.14 |
|  | CCF | 1 | 0.15 | 0.14 |
| 2 | CF | 0.04 | 0.03 | 0.08 |
|  | CCF | 0.04 | 0.0045 | 0.0112 |
| 3 | CF | 0.04 | 0.03 | 0.09 |
|  | CCF | 0.0016 | 0.000135 | 0.001008 |

The information from the players' weekly popularity time series from Eq 3 was summarised in six popularity indicators (PIs) that can be used in cross-sectional studies: the *first principal component (PC1)*, *mean*, *median*, *maximum*, *minimum*, and *standard deviation*. (The principal component analysis was performed using the popularity information for each week of the players. The information stored in the first principal component explained a little more than 50% of the variability.)

## Materials and methods

This section presents the database used to carry out the study and describes the statistical methods employed in the predictive analysis. The R software package was used to analyse the database [33]. It provides high-quality statistical and graphical methods. RStudio [34], an integrated development environment (IDE), was also used to write the programs in R.

### Data

The initial database comprised 1,428 players, corresponding to the players who competed in LaLiga, the Premier League, Bundesliga, Serie A, and Ligue 1 during the 2018–2019 season, and 43 explanatory variables grouped into four types: the *player characteristics*, *player performance*, *labour and club characteristics of the players*, and *player popularity* (see Table 3). Note that our primary data source, as well as all code associated with this project is available at https://github.com/Malagon-Selma/Market-Value-BIG5-2018-2019. In particular, the data was collected through the Internet websites of Google Trends *(trends.google.es)*, WhoScored

**Table 3. Dependent and explanatory variables for estimating players' market value.**

| Indicator | Variable and abbreviation |
|---|---|
| Dependent variable | Transfermarkt's market value (training) andTransfer fees (model error) |
| Player characteristics | Position (P), Footedness (FT), Age, Height (H), and Continent (Cnt) |
| Player performance | Playing time (PT), Aerial duel accuracy (ADA), Tackle accuracy (TA), Shots intercepted (SI), Fouls (F), Yellow cards (YC), Red cards (RC), Goals (Gls), Shots (S), Shot accuracy (SA), Assists (A), Dribbles (Dr), Crosses (Cr), Corners (C), Passing accuracy (PA), Short-pass accuracy (SPA), Long-pass accuracy (LPA), Key passes (KP, passes that create shots for teammates), Progressive passes (PP, passes that move the ball to the opponent's goal), Deep passes (DP, passes into the space between defenders), Passes in the penalty area (PPA), Passes in the last quarter of the opponent's half (PLO), and Free kicks (FK) |
| Labour and club characteristics of the players | Contract (Ct) and League |
| Popularity indicators | GTA, PC1, Mean (Mn), Median (Mdn), Maximum (Max), Minimum (Min), and Standard deviation (Sd) |

*(whoscored.com)*, FBref *(fbref.com)*, and Transfermarkt *(transfermarkt.es)*. All the data were collected according to the Terms of Use and Service of the source websites.

Regarding the variables related to player performance, the game actions are measured as the average number of actions per game, calculated at the end of the season. In addition, the explanatory variables were standardised by subtracting the mean from the data. The models were trained using, as the dependent variable, the market values of 1,235 unsigned players (training set), while the model error was assessed using the transfer fees of the 193 players signed during the summer market (test set). Therefore, the models were trained using the market values estimated by Transfermarkt as the dependent variable, although the model error was computed using the transfer fees of the players that were signed. Müller et al. [7] previously carried out this procedure.

Note that in the case of the PIs, each player's GTN time series was calculated by considering the number of times each player had been searched worldwide in the category 'Web Search', so any search by name, in the image, news, or shopping categories, or on YouTube was included. As the study concerned players who competed throughout the 2018–2019 season, the period from 17 May 2018 (the date when the summer market opens in England, the first league to start) to 26 May 2019 (the date when the Italian league ends, as it is the last league to end) was chosen for the analysis. Consequently, 54 values (popularity per week) were stored for each player and used to construct the popularity indicators.

Finally, the league and continent variables were added to the models as categorical variables. Thus, players were classified according to the league in which their home team competed (LaLiga, Premier League, Serie A, Bundesliga, and Ligue 1). The second categorical variable represented the continent of birth of each player (Europe, Asia, Australia, Africa, North America, and South America).

## Methods

The multivariate methods selected to train and test the models were the multiple linear regression (MLR) [35], random forest (RF) [36], and gradient boosting machine (GBM) [37, 38] methods. MLR is a classical method that is used as a benchmark, and the RF and GBM methods were selected due to the good results they have achieved in most of the research fields in which they have been applied. In addition, previous researchers have used these machine learning techniques to determine the market value using the RF [10] and gradient boosting, namely XGBoost [39–41]. Note that both the RF and GBM make it possible to analyse the importance of a variable.

**Multiple linear regression (MLR).**   Multiple linear regression (MLR) is a classical method in which a dependent variable ($Y$) is predicted based on a linear combination of explanatory variables ($X_1, \ldots, X_i$). In addition, MLR allows us to determine the effect of the explanatory variables on the dependent variable (statistically significant and non-significant variables). However, before performing MLR, it was necessary to ensure that there was no multicollinearity between the explanatory variables. The variance inflation factor (VIF) quantifies the collinearity between these variables. A high VIF indicates a high collinearity. Thus, `vif_function` [42] was used to remove those variables with a VIF higher than 2.5 [43, 44]. Once the multicollinearity was controlled, the stepwise algorithm selected the most relevant variables in the fitted linear model according to the Akaike information criterion (AIC) [45].

**Random forest (RF).**   The random forest (RF) is a predictive supervised learning method that uses the bagging technique [46] to improve the prediction capacity by combining independently constructed trees. The process starts with the random sampling with replacement of $n$ observations from the training database. The unsampled observations are called out-of-bag

(OOB) observations. Then, a decision tree with a subset of randomly selected variables for each division is created and used to predict OOB observations, and the prediction error is measured. The 'mtry' parameter determines the number of variables selected for each split. This parameter should be optimised to improve the predictive capacity of the model. Additionally, the RF allows us to determine the importance of the variables in the regression model [36].

Liaw and Wiener [47] implemented IncMSE% and IncNodePurity in their R-package. IncMSE% measures the increase in the mean squared error (MSE) in the OOB observations when one variable's values are permuted in the training dataset while the others remain unchanged (the greater the increase in the MSE, the greater the importance of the variable). IncNodePurity measures the increase in the purity each time a node is split for a given variable. In regression, the node impurity is related to the MSE: an increase in the node purity implies a decrease in the MSE. IncNodePurity is calculated for each tree and finally normalised by the number of trees in which the variable has been used (the greater the increase in the purity, the greater the importance of the variable [36]).

**Gradient boosting machine (GBM).**   The gradient boosting machine (GBM) is another predictive supervised learning method used for both regression and classification, like the RF. Impulse methods such as the GBM arose from the question of whether a set of weak classifiers can lead to a robust model [48, 49]. The GBM creates several predictors in sequence (set of trees) that are dependent on each other, shallow, and weak, i.e. the error rate is barely better than guessing. The main idea is to train models (trees) sequentially so that each model fits the errors of the previous model (tree).

The algorithm begins by fitting a tree to the data (Eq 4), where $x$ represents the explanatory variables and $y$ represents the response variable:

$$y = f_1(x). \tag{4}$$

The following tree fits the *pseudo-residuals* of the previous tree (it is not intended to predict the response variable; instead, it is meant to predict the pseudo-residuals). Next, a new tree calculates the pseudo-residuals of the two previous models combined. This process continues until some criterion (in our case, cross-validation) indicates that the process should stop. In this way, each model or tree provides a better prediction than the previous tree. Note that each new tree will predict the pseudo-residuals of the previous tree. Therefore, the division criterion is usually different in each new tree.

One of the challenges of the GBM is that it has a wide range of hyperparameters that must be tuned. For our analysis, the gbm R-package [50] was used to optimise the hyperparameters. The hyperparameters were the number of trees (total number of trees to adjust), depth of the trees (number of divisions in each tree), and learning rate (a value between 0 and 1 used to avoid overfitting).

Finally, the contributions of the variables are obtained similarly to the importance in the RF (in the case of the GBM, the entire training dataset is used, not just the OOB observations). Specifically, predictor variables are randomly permuted (one variable at a time), and the decrease in the predictive performance is calculated. This value is averaged over all trees and for each variable. The greater the average decrease in the precision, the greater the contribution of the variable.

## Models

Three different models were considered to determine whether the proposed PIs significantly influence the market value and whether they improve the prediction of a player's transfer fee compared to the GTA alone. In model 1 (Eq 5), which served as the baseline model, the

transfer fee for each player is explained by the *player characteristics*, *player performance*, and *labour and club characteristics of the players*. Model 2 (Eq 6) added the *GTA* to the baseline model, whereas model 3 (Eq 7), instead of the *GTA*, included the *PIs* calculated using the method proposed for the construction of popularity indicators. Note that the specific variables included in each model are listed in Table 3:

$$Transferfee_i = f(characteristics_i, performance_i, labour/club_i), \tag{5}$$

$$Transferfee_i = f(characteristics_i, performance_i, labour/club_i, GTA_i), \tag{6}$$

$$Transferfee_i = f(characteristics_i, performance_i, labour/club_i, PIs_i). \tag{7}$$

## Repeated *k*-fold cross-validation

Repeated *k*-fold cross-validation (CV) was used to optimise the hyperparameters of the training set for the three methods (MLR, RF, and GBM) using the `caret` R-package [51]. This method randomly divides the database into *k* subgroups (in this case, *k* = 5 was used). Then, the training set is created with *k*−1 subgroups, and the validation set (to validate the model) is made with the remaining subgroup. This process validates the model on non-overlapping datasets in each iteration, and the process concludes after all individuals have been in the validation set once (and only once) [52].

Additionally, *k*-fold CV is repeated (in this case, CV was repeated 5 times).

After training the models and optimising the hyperparameters, the model performance is evaluated using the transfer fees of 193 players who were not used to build the model. The root mean square error (RMSE) is typically used to determine the difference between the observed values and the values predicted by the regression models, and it can be calculated as follows:

$$RMSE = \sqrt{\sum_{i=1}^{n} \frac{(\hat{y}_i - y_i)^2}{n}}, \tag{8}$$

where $y_i$ represents the observed values and $\hat{y}i$ represents the corresponding predicted values for each observation $i = 1, \ldots, n$. Therefore, the higher the RMSE (Eq 8), the worse the fit of the prediction model to the data.

## Results

Machine learning culture was applied to validate and compare the models. Following Breiman [53], repeated k-fold cross-validation was used to avoid overfitting, and then, with the final selected model, we predicted the market values of the test set. Table 4 shows the goodness of fit calculated using the RMSE, mean absolute error (MAE), and $R^2$ on the training set.

According to Table 4, model 3 is the model that best fits the data for all three methods, with the GBM being the method with the lowest RMSE (9,442,425) and MAE (5,409,197) and the highest $R^2$ (0.797). Table 5 shows the performance of each method and model according to the RMSE on the test set.

Table 6 shows the estimated coefficients for the three models after the collinearity is removed and the stepwise algorithm using the AIC is applied as part of MLR.

According to Table 5, there is no difference in the performance of models 1 and 2 when they are fitted using the MLR method. This result is explained by the variable selection process

**Table 4. Goodness-of-fit statistics for the three final models fitted using all three methods to the training set (€).**

| | | RMSE | MAE | R$^2$ |
|---|---|---|---|---|
| Model 1 | MLR | 14, 647, 204 | 9, 907, 675 | 0.509 |
| | RF | 14, 867, 996 | 9, 310, 879 | 0.514 |
| | GBM | 13, 432, 903 | 8, 107, 707 | 0.586 |
| Model 2 | MLR | 14, 666, 566 | 9, 925, 326 | 0.507 |
| | RF | 14, 844, 071 | 9, 321, 096 | 0.519 |
| | GBM | 13, 494, 724 | 8, 188, 325 | 0.589 |
| Model 3 | MLR | 14, 388, 569 | 9, 446, 940 | 0.534 |
| | RF | 10, 522, 169 | 5, 877, 046 | 0.755 |
| | GBM | 9, 442, 425 | 5, 409, 197 | 0.797 |

**Table 5. RMSE for all three models fitted using all three methods to the test set (€).**

| | MLR | RF | GBM |
|---|---|---|---|
| Model 1 | 16, 476, 016 | 16, 248, 118 | 18, 000, 361 |
| Model 2 | 16, 476, 016 | 16, 297, 149 | 16, 852, 699 |
| Model 3 | 15, 763, 270 | 12, 076, 227 | 11, 507, 727 |

since, in model 2, the AIC caused the *GTA* variable to be discarded (see Table 6). Thus, model 1 and model 2 use the same variables. On the contrary, in model 3, the PI *Min* is included in the base model and is statistically significant. Moreover, the positive coefficients of *Min* show that increasing this popularity indicator will increase the market value (see Table 6).

In the case of the RF method, Table 5 shows a slight difference between models 1 and 2, with model 1 being more accurate than model 2. In contrast to the RF method, model 2 gives a more accurate result than model 1 when they are fitted using the GBM method. Additionally, as in MLR (Table 6), the variable *GTA* is not among the most important variables in model 2, while some PIs are important variables for the market value (see S1–S3 Figs). Furthermore, Table 5 confirms what is shown in Table 4 since the GBM RMSE is the lowest RMSE for model 3.

Therefore, it is possible to conclude that using additional topics to normalise the time series provided by GT, i.e. using the GTN, and creating PIs that summarise the popularity of players over time is a helpful way to decrease the errors in the predicted transfer fees. Thus, using the PIs instead of the *GTA* caused the RMSE to decrease by €5,344,972, €4,220,922, and €712,745 for the GBM, RF, and MLR methods, respectively. In order to improve the predictions and learn about the PIs that have the most impact on the market value of a football player, an analysis of the variables that most contribute to the prediction was carried out. Specifically, the variables selected when the GBM method was used to fit model 3 were studied since this combination of variables yielded the best predictions. The variables selected by the other models and methods are shown in the Suppoting information.

Fig 1 is a bar plot with the names of the variables (on the *y*-axis) and their mean contributions (on the *x*-axis), which were calculated using the *gbm* R-package [50].

Fig 1 shows that the most crucial variable for predicting the market value of a player signed during the summer market of the 2018/2019 season is the popularity indicator *Min*, which contains information about the week in which the player was searched the least. Note that *Min* was the most important variable in the case of the GBM (Fig 1) and RF (S2 Fig) methods, and it was statistically significant in the case of MLR (Table 6).

**Table 6. Coefficients of the statistically significant variables for the three models fitted to the data using the MLR method.**

**Dependent variable: Market value**

| Variables: | Model 1 | Model 2 | Model 3 |
|---|---|---|---|
| Intercept | 16,368,678*** | 16, 368, 678*** | 16, 396, 118*** |
| Age | -1,047,445*** | −1, 047, 445*** | −1, 110, 551*** |
| Ct | 2, 195, 638*** | 2, 195, 638*** | 2, 321, 530*** |
| PT | 5, 403*** | 5, 403*** | 5, 307*** |
| SI | −5, 562, 808* | −5, 562, 808* | −5, 741, 650* |
| F | −2, 205, 030** | −2, 205, 030** | −1, 856, 055* |
| Gls | 53, 090, 883*** | 53, 090, 883*** | 45, 161, 546*** |
| A | 22, 990, 721*** | 22, 990, 721*** | 20, 386, 278*** |
| SPA | 1, 080, 304*** | 1, 080, 304*** | 1, 009, 181*** |
| LPA | 78, 156* | 78, 156* | 79, 372* |
| DP | 9, 499, 431*** | 9, 499, 431*** | 8, 202, 002*** |
| FK | −4, 736, 462*** | −4, 736, 462*** | −4, 598, 529*** |
| ADA | 115, 128** | 115, 128** | |
| Dr | 757, 173** | 757, 173** | |
| H | | | 111, 469 |
| TA | | | −31, 137 |
| Cr | | | 696, 646. |
| Ligue 1 | −5, 368, 052*** | −5, 368, 052*** | −5, 303, 768*** |
| Premier League | 7, 712, 282*** | 7, 712, 282*** | 7, 509, 600*** |
| Serie A | −2, 870, 454* | −2, 870, 454* | −3, 034, 731*** |
| South America | 3, 715, 759** | 3, 715, 759** | 3, 611, 187** |
| Min | | | 10, 047, 141*** |

Significance codes:. $p < 0.1$

\* $p < 0.05$

\*\* $p < 0.01$

\*\*\* $p < 0.001$

The high contribution of *Min* could be explained by the fact that this variable contains information on the number of 'actual' or 'unconditional' fans since *Min* provides information on the popularity of players when they are less popular. Therefore, this variable is not altered by a temporal increase in the popularity of the players due to outstanding playing actions (positive or negative). Furthermore, Fig 1 shows that *Max* is the popularity indicator with the lowest contribution to the market value prediction. This result aligns with the previous result, as *Max* and *Min* contain opposite types of information. While *Min* only considers 'unconditional' fans, *Max* also includes the number of Google searches performed by 'conditional' fans, i.e. football fans who only consider the player after a prominent action. Thus, *Min* is a robust indicator of popularity.

In the case of the variables related to player characteristics, the three models are in line with the results of previous research [7, 9, 13, 20, 21], which found that *Age* is a variable with a high contribution to the market value, whereas *H* and *FT* do not make essential contributions. In the case of the *P* variable, our results differ from those of previous studies [6, 7, 10], which found that this variable was statistically significant. None of our models found this variable to be important or statistically significant in predicting the players' market value. As for the

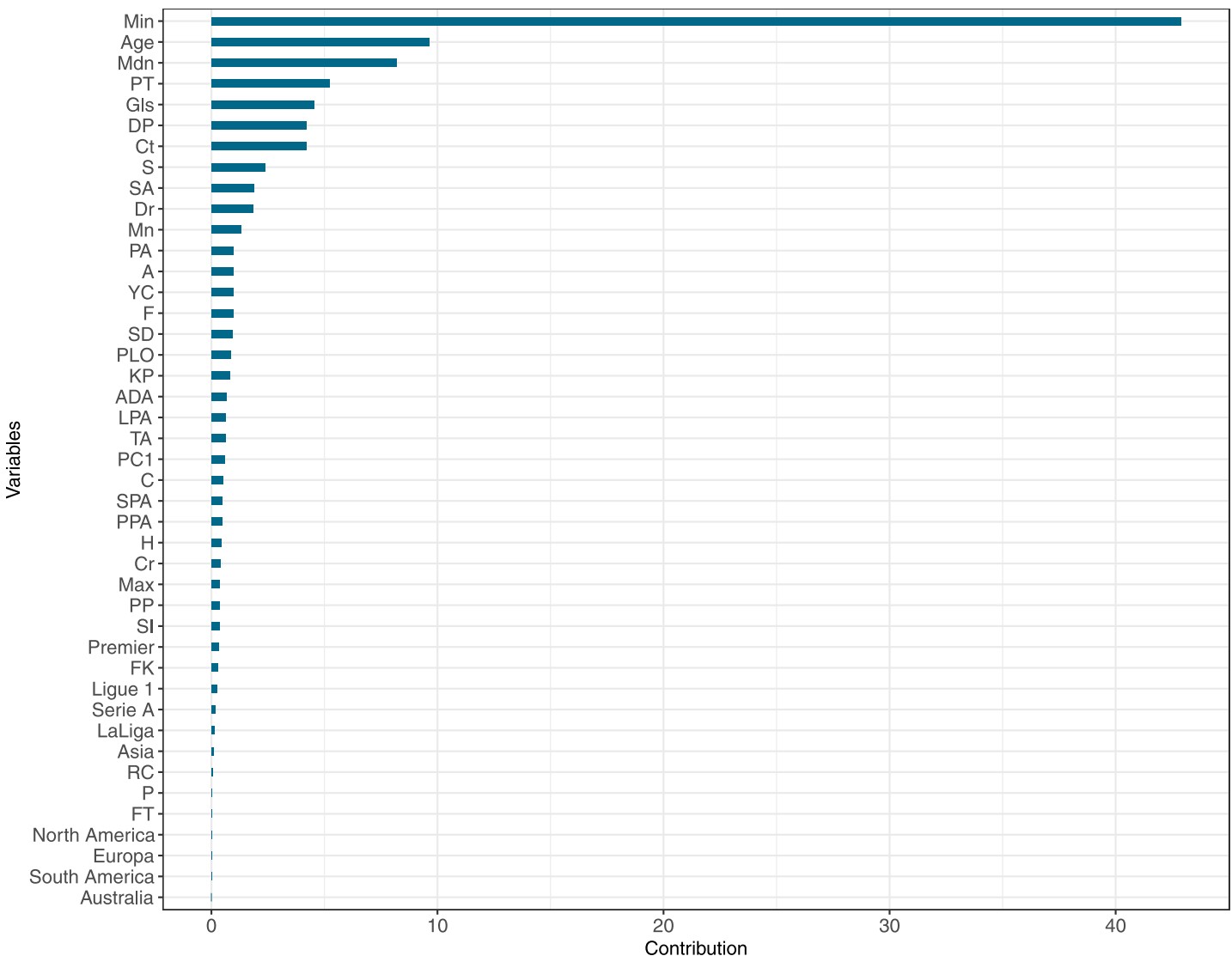

**Fig 1. Contributions of the variables of model 3 when it is fitted to the data using the GBM method.**

continent variable, which stores information on the nationalities of the players, the methods used to carry out the prediction differ. In the case of MLR, the results showed that players born in South America are overvalued, as the coefficient was positive and statistically significant (see Table 6). This result is in line with the work of Pedace [19]. However, neither the RF method nor the GBM method found this variable to be important or contributive in predicting players' market value (see Fig 1 and S2 Fig).

In terms of the variables related to player performance, this study agrees with previous research since it highlights the importance (Fig 1 and S2 Fig) and statistical significance (Table 6) of variables such as *PT* [7, 10, 18], *Gls* [7, 10, 13], *Dr* [7, 9, 10], and *S* [9]. Note that *S* was not statistically significant in any model fitted by the MLR method. It is highlighted that although the variable *DP* has not been analysed in previous research, it was found to be either important (Fig 1 and S2 Fig) or statistically significant (Table 6) for all three methods.

As far as the labour and club characteristics of the players are concerned, the results for the *Ct* variable (see Table 6, Fig 1 and S2 Fig) are largely in line with those of previous researchers [18, 24–26], who pointed out that the contract's duration influences the players' market value and in fact increases this value. Finally, in terms of players' leagues, the MLR method indicates that players competing in Ligue 1 and Serie A are undervalued. In contrast, the opposite is true for players in the Premier League (Table 6). This finding agrees with the results of previous researchers, who found statistically significant differences between players competing in the Premier League and players competing in other leagues [20, 21]. However, in the case of the RF and GBM methods, even though in model 2 (S1–S3 Figs), competing in the Premier League contributes to predicting the market value of the players, this effect is not found in model 3 (Fig 1 and S2 Fig).

## Discussion

Although the effect of players' popularity on the market value has been explored [7, 10, 12, 13, 15], this study proposes new ways to use GT data to measure players' popularity by calculating PIs. GT is a valuable tool that contains information on Google searches by name, in the image, news, and shopping categories, or on YouTube. Therefore, while previous researchers have used data from different web sources (e.g. *the number of visits to Wikipedia*, *YouTube views*, or *Google visits*), GT summarises all this information in a single value that provides a global view of the popularity of the players. In addition, while most of the variables listed above store information at a specific point in time, GT provides information over a time range. Another problem that GT avoids is missing data because, unlike variables such as the *number of followers on Instagram*, *Facebook*, or *Twitter* that depend on players having social media accounts, GT considers any appearance on Google. Finally, with GT, we can mitigate the problem of multicollinearity, as all popularity information is stored in a single variable.

To the best of our knowledge, this is the first study to propose the construction of PIs by adding additional topics to rescale the GT information and classify players according to their 'popularity layer' to facilitate comparisons between players with very unequal popularity levels. In addition, to determine if the method proposed by Müller et al. [7] could be improved, this paper analysed the performance of the methods when the *GTA* (model 2), which stores the GT information that was used by Müller et al. [7], and the PIs calculated according to the proposed method (model 3) are used.

The results of model 2 are in line with those of Müller et al. [7], since the AIC, which selects the most relevant variables for fitting using MLR, caused the *GTA* variable to be discarded, which means that each player's individual GT time series, as measured by Müller et al. [7], does not contain information about that player's popularity compared to the popularity of another player, possibly due to normalisation. On the contrary, model 3 showed the significant influence of the PIs on the market value (Fig 1); specifically, the variable *Min* was the most influential. This fact highlights that the proposed methodology is helpful for correctly obtaining GT information and avoiding problems due to normalisation.

Additionally, the results presented in Table 5 show that the PIs improve the prediction of transfer fees since the model error of the GBM method using PIs was €11,507,727, while when the *GTA* was used, it was €16,852,699.

The performance assessment of the methods and models confirms the helpfulness of including the proposed PIs when it comes to predicting transfer fees. The model 3 error for the GBM method (the lowest error) using PIs was €11,507,727, while when no measure of player popularity was used, it was €18,000,361 (Table 5). Thus, when PIs were used, the model error decreased by €6,492,634; this is a large decrease considering the high impact of transfer fees in football today.

The results obtained in this study agree with the results of previous investigations, which indicated that the variables related to the players' popularity contribute to predicting the players' market value [7, 12, 13, 15]. However, many of the previous studies that used popularity variables did not quantify the improvement in the prediction of transfer fees that was produced by the addition of popularity variables.

Garcia-del Barrio and Pujol [12] created two models, but they used the *R*-squared goodness-of-fit value to measure model improvement. Müller et al. [7] developed four multilevel regression models and compared the RMSE of their best model with Transfermarkt's RMSE.

Similarly, Singh and Lamba [10] calculated the RMSE of the log-transformed dependent variable, but they did not provide an error measure regarding the market value. Unlike this work, Hofmann et al. [15] did not study the specific effect of popularity variables on the market value; instead, they studied the effect of the brand image on player popularity. However, their study also corroborated the statistically significant effect of popularity variables on the market value. In summary, although most researchers who used popularity variables found them statistically significant, most did not study or quantify the specific effect of popularity variables on the prediction of transfer fees. Therefore, this study contributes to the literature by providing a novel method and a detailed analysis of the impact of popularity measures on the prediction of transfer fees.

The results emphasise the importance of players' systematic brand images on the market value and, therefore, the revenue of clubs. Therefore, clubs and agents should focus on maintaining or improving the positions of the players and establishing mechanisms to increase their popularity. However, it should not be forgotten that performance-based attributes have direct and indirect effects on player popularity, and not the other way around [15]. Thus, there are two key conclusions: a) players have to enhance their performance attributes, which will support their media-related attributes, and b) managers and clubs have to apply successful strategies to improve or maintain the brand images of players.

## Conclusion

This work presents a novel way of calculating popularity indicators (PIs) using Google Trends (GT) time series that involves requesting the time series of two football players simultaneously. This work contributes to the study of the effect of football players' popularity on the market value in order to use players' popularity to predict transfer fees. The analyses were conducted on a sample of 1,428 players from the 'Big Five' European football leagues over a period of one playing season (2018–2019). We found that by rescaling GT time series and summarising them using PIs, we can overcome the limitations of previous studies.

The central conclusion of this analysis is that the proposed PIs improved the prediction of transfer fees. Moreover, we have confirmed the marked impact of popularity on players' market value. Specifically, *Min* is a robust reflection of popularity, with an important and significant effect on the market value; it improves the prediction of transfer fees.

In addition, we would like to note that although this paper only obtains PIs for football players, the method can be adapted to compare the popularity levels of any search terms about any topic. Therefore, this method has a huge range of potential applications.

Despite its potential contributions to the literature, our study has some limitations. First, although our goal of demonstrating the usefulness of PIs in predicting transfer fees has been achieved, our prediction error remains somewhat large. This is (likely) due to the fact that our model uses data from a single season. Thus, future research should involve performing similar analyses that consider a greater number of seasons. Second, due to the difficulty of finding estimates of the market value, we have trained the models with estimates made by Transfermarkt.

Therefore, future researchers are encouraged to use historical data-driven transfer fees to create a database that is large enough to use to train and test the model so that the model could be trained and tested on actual transfer fees. Third, recent studies on GT have demonstrated that it may provide inconsistent data, which affects the accuracy of the GT measurements. Thus, future research should apply corrective measures to avoid potential bias problems. Future research could focus on creating position-specific models in the manner of Majewski [22], since certain game actions, such as goals or saves, may have greater significance for forwards or goalkeepers, respectively. Finally, another interesting variable that future research should consider is player injuries. Although this information has not been widely used so far, it can influence the players' performance variables, especially if the injury occurs at the beginning of the season. Therefore, it could also impact the players' market value.

To provide a complete view of the impact of players' popularity on their market value, future research should also include longitudinal data to capture both economic growth and crisis periods, as club revenues could significantly influence market values [54].

## Supporting information

**S1 Fig. Importance of the variables of model 2 when it is fitted to the data using the RF method.**
(EPS)

**S2 Fig. Importance of the variables of model 3 when it is fitted to the data using the RF method.**
(EPS)

**S3 Fig. Contributions of the variables of model 2 when it is fitted to the data using the GBM method.**
(EPS)

## Author Contributions

**Writing – original draft:** Pilar Malagón-Selma.

**Writing – review & editing:** Ana Debón, Josep Domenech.

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
