## [Decision Letter · Decision Letter 0]

13 Mar 2023

PONE-D-22-31349Measuring the popularity of football players with Google TrendsPLOS ONE

Dear Dr. Malagón-Selma,

Thank you for submitting your manuscript to PLOS ONE. After careful consideration, we feel that it has merit but does not fully meet PLOS ONE’s publication criteria as it currently stands. Therefore, we invite you to submit a revised version of the manuscript that addresses the points raised during the review process.

We look forward to receiving your revised manuscript.

Kind regards,

Prof. Dr. Florian Follert

Academic Editor

PLOS ONE

Journal Requirements:

2. In your Methods section, please include additional information about your dataset and ensure that you have included a statement specifying whether the collection and analysis method complied with the terms and conditions for the source of the data.

P.M-S. Economic support through the FPI-UPV scholarship (PAID-01-19) to the Universitat Politècnica de València. http://www.upv.es/entidades/VINV/indexc.html

P.M-S, A.D., and J.D. Grant PID2019-107765RB-I00, funded by MCIN/AEI/10.13039/501100011033.

Reviewers' comments:

Reviewer's Responses to Questions

**Comments to the Author**

1. Is the manuscript technically sound, and do the data support the conclusions?

Reviewer #1: Partly

Reviewer #2: Yes

2. Has the statistical analysis been performed appropriately and rigorously? 

Reviewer #1: No

Reviewer #2: Yes

3. Have the authors made all data underlying the findings in their manuscript fully available?

Reviewer #1: Yes

Reviewer #2: No

4. Is the manuscript presented in an intelligible fashion and written in standard English?

Reviewer #1: Yes

Reviewer #2: Yes

5. Review Comments to the Author

Reviewer #1: Generally, my opinion is positive.

My main suggestion is to supply econometric modeling by full model verification.

During the reading of the work I find some doubts:

1) the data are not heterogenous - it could cause of the weakness of models;

2) there is a lack of measures of goodness-to-fit of models used;

3) joining in one group of footballers playing on different positions on the pitch is not correct (different roles on the pitch) - this problem [look work of Majewski, 2016] influences the popularity of the player (goalkeepers or defenders have not to be on the top of best scorers list - other features are significant for the goalkeeper and other for the striker. The such limitation should be taken into account, because different features may determine the popularity of players depending on the game's position.

4) The authors made their research on 1428 players from BIG-5 leagues from the years 2018-2019. There is no information about the type of time series used (daily, weekly or monthly). Structurally, the five leagues used in the research differ from each other. So, there could exist also heterogeneity in the field of leagues. In my opinion, the results obtained for such heterogeneous objects are incorrect. Please, explain this doubt. Maybe there could be used some dummy variables representing different leagues? Maybe it could be done also for years.

5) page 11/26 - the predictive analysis should be run after estimation and verification - there is a lack of verification of models.

6) There is a lack of results for RF analysis and the GBM model. What do models look like? Which variables are included? Which variables are statistically significant?

7) Authors run the discussion only with one research - authors show differences between their own and corresponding research. Is there any other research?

Despite the fact the authors would like to enter some novelty in the research in this field, I think the presentation of models without full verification is wrong.

Reviewer #2: Dear author(s):

Thank you for the opportunity to read your manuscript "Measuring the popularity of football players with Google Trends". The study uses a dataset comprised of 1428 football players from five European leagues during the 2018-2019 season and calculates popularity indicators to measure the popularity of these football players and the contribution of popularity to predict players’ market value.

Please find my detailed comments below to aid you in the further development of your study.

Abstract

• The abstract should be more straight forward and could need a bit of restructuring to make it clearer and to have a better understanding at first glimpse. What is the research question, what are the methods used, what are the results and findings, as well as the implications?

Introduction

• You write that “The sports industry is a scenario where most interests converge, from political to public, increased by communication media and information.” I am not sure I understand this sentence. Could you reformulate it please?

• What exactly do the authors refer to when mentioning “income”? Club revenues, Gross income, net income? Player income?

• You state “Football club revenues are often closely related to teams' investment in their main assets: the players”, as has been previously pinpointed by Frick (2007). But what do you exactly mean with this sentence, that clubs with higher revenues are able to purchase players that have higher transfer fees?

• Since you study a probable effect of “popularity” on “market value”, you should first of all properly assess the two key variables, i.e. popularity and market value. Popularity is quite detailed described in section 1.2. As of “market value” you cite Herm et al. (2014) with “an estimate of the amount of money that a club would be willing to pay for an athlete contract, regardless of an actual transaction”. However, this definition has been criticized in the past. What “club” do Herm et al. refer to? Does the term “market” not rather suggest that we are not considering one club but several clubs or several stakeholders? How do we know what a single club is willing to pay if they do not communicate it? And in negotiation process or even after, it is highly unlikely that a club would communicate the prize they would have been willing to pay. Given that in your study you do not ask one or several clubs for their willingness, but rather obtain your market values from transfermarkt, you should be consistent here. A discussion of transfer and market value has been provided in a recent working paper by Franceschi et el. (What can football economics literature learn from subjective value theory?). They also provide an overview of alternative definitions in their appendix.

Background

• The section is rather short and the summary of previous research performed in the three described parts (Player characteristics, Player performance, and Player popularity) is not elaborately enough and too unspecific. I suggest you explain the findings in much more detail and the partly conflicting results of different existing studies. The results are not as unitary as you make it sound. Please have a look at Franceschi et el. (2023) for a recent literature review on football player valuations.

Ref

Franceschi, M., Brocard, J. F., Follert, F., & Gouguet, J. J. (2023). Determinants of football players’ valuation: A systematic review. Journal of Economic Surveys.

• You write that “Researchers have identified three principal groups of variables for analysing a player's market value: Player characteristics, Player performance, and Player popularity.” Since you site Frick (2007) you are aware that there are a few more factors. You could further add a section for other influencing factors that cannot be summarized under these three categories but that have been found to influence market value in the past, such as current market conditions, current and future clubs, leagues of the selling and purchasing club, the nationality of a player – in fact, some of these factors could be factors that influence market value via your variable popularity. A player that has been playing for a relatively short time at a club outside Europe and is suddenly purchased by a top club receives a big boost in terms of popularity (latest example Endrick from Palmeiras who will be transferred to Real Madrid in 2024).

• In the section on Player characteristics, you refer to previous analyses that have found contract length to influence performance. What do you want to tell the reader here? Is remaining contract length a player characteristic? And why do you write about performance as an endogenous variable here?

• You write that “while the characteristics and performance of players have helped predict their transfer fees, they are not perfect measures of a football player's market value, so complementing them with popularity data could offer a better idea of a player's valuation.” This assertion falls somewhat from the sky and you do not refer to previous research nor have a comprehensible rationale for this claim. Please elaborate.

• You write that “…the brand image is also sold or exchanged with the player…it is necessary to study the specific effect of popularity on the transfer fee.”

The concept of image rights, as understood today, stems from the so-called “right of publicity” (Al-Ameen, 2017). However, the ownership and control of image rights, which can be exploited through sponsorship, endorsement and merchandising, is not subject to legal rules on allocation of rights and responsibilities but is a mere question of negotiations between clubs and players (Haynes, 2007). It is common practice that clubs enter into employment contracts with their athletes that require them to cede the use of their brand or image for marketing and commercial activities (Hewison, 2001). While a few powerful football club such as Arsenal, Manchester United and Real Madrid demand full control over players’ image rights, other clubs are just content to get a minimal share of the proceeds from the commercial exploitation of the player’s image rights (Al-Ameen, 2017). Star players in top leagues strive to circumvent the image right obligations by, often successfully, leveraging their position to obtain advantageous conditions, whereas average players are often denied this privilege.

Ref.

Al-Ameen, H. A. (2017). Image right clauses in football contracts: masterstroke for mutual success? Intellectual Property Rights, 5(1).

Hewison, J. (2001, 07.12.2001). Legal teams take to the football field to thrash out standard player's contract. Retrieved 10.08.2022 from https://www.lawgazette.co.uk/news/legal-teams-take-to-the-football-field-to-thrash-out-standard-players-contract/35706.article

• Some rather recent articles using machine learning technics in determining market value have not been considered at all, such as

McHale, I. G., & Holmes, B. (2022). Estimating transfer fees of professional footballers using advanced performance metrics and machine learning. European Journal of Operational Research.

Yigit, A. T., Samak, B., & Kaya, T. (2020). An XGBoost-lasso ensemble modeling approach to football player value assessment. Journal of Intelligent & Fuzzy Systems, 39(5), 6303-6314.

Zhang, D., & Kang, C. (2021, April). Players’ Value Prediction Based on Machine Learning Method. In Journal of Physics: Conference Series (Vol. 1865, No. 4, p. 042016). IOP Publishing.

Materials and methods

• What was the reason you only included players of the big 5 leagues? Maybe you could perform the analysis by additionally using players of other popular leagues such as the Eredivisie, German Bundesliga 2, Portuguese Primeira Liga, and Belgium First Division where there is enough information available and the players are known, thus do not cause problems with market value. Additionally, these are the leagues where a lot of players of the Big 5 leagues are bought from. This way you would have a bigger dataset than that of 1428 players, which is a rather small dataset (as you pointed out yourself in your concluding remarks) – sample size is important, especially in a machine learning context and the degrees of freedoms you have.

• Do you have examples of other studies where the model has been trained with one data set (in your case market values) and then tested with another dataset (transfer fees)?

• As for the dependent and explanatory variables used, there might be the need to add a variable indicating player injuries. Injuries are very common in the game of football, and while short to midterm injuries have little impact on market values, they certainly impact virtually all player performance variables.

• Previous research has found evidence for customer and employer-based discrimination based on nationality in the case of football (Pedace, 2008). Players from highly populated and football enthusiastic countries such as Mexico and Brazil might have distorted popularity figures when comparing these figures to their market values. One such case was James Rodriguez when playing for Bayern Munich between 2017 and 2019. In your study you do not control for nationality or home population

Ref

Pedace, R. (2008). Earnings, performance, and nationality discrimination in a highly competitive labor market as an analysis of the English professional soccer league. Journal of Sports Economics, 9(2), 115-140.

• You used 5 different leagues and the market values as provided by transfermarkt with two different currencies, EUR and GBP. Are you aware that the transfermarkt data calculates past transactions at current exchange rate? How do you account for these exchange rate discrepancies?

• You might want to explain why you have chosen random forest and gradient boosting machine among the many machine learning models that are used as of today.

• Table 4 needs a makeover, it is hard to read and interpret. Having said that, the results of the MLR are not discussed. There is only a comparison between the models.

Discussion

• How do the results of the paper contrast to other studies that examined the impact of player popularity on transfer fees?

• Please detail the implications of this paper and your results for management

6. PLOS authors have the option to publish the peer review history of their article (what does this mean?). If published, this will include your full peer review and any attached files.

Reviewer #1: No

Reviewer #2: No

---

## [Author Response · Author response to Decision Letter 0]

6 Jun 2023

We enclose the revised version of our article Measuring the popularity of football players with Google Trends (ref. PONE-D-22-31349) in which we have incorporated the referees’ changes and suggestions. We would like to thank the Reviewers for their insightful comments that have allowed us to really improve the original version of the paper.

Next, the changes made in the paper following the referee's recommendations are detailed and explained. 

In addition, we also enclose written responses for the referees in which we try to give an answer to each of the points made in their reports. 

Please note that the page numbers referenced in this document match those in the document titled: "Revised Manuscript with Track Changes"

Yours sincerely,

Pilar Malagón

 

Response to referee 1:

We appreciate the changes and suggestions indicated and your detailed reading of the paper. The answers to the comments made are detailed below, and the suggested changes are highlighted in the text.

1) The data are not heterogenous - it could cause of the weakness of models.

This comment is very interesting. While this could be a potential limitation, we feel that the heterogeneity in our data is similar to previous research works which used similar data for predicting the players' market value [5, 8,11, 13].

References:

[5] Müller O, Simons A, Weinmann M. Beyond crowd judgments: data-driven estimation of market value in association football. Eur. J. Oper. Res. 2017 Dec;263(2):611-624.

[8] Singh P, Lamba PS. Influence of crowdsourcing, popularity and previous year statistics in market value estimation of football players. J. Discrete Math. Sci. Cryptogr. 2019 Feb;22(2):113-126.

[11] Herm S, Callsen-Bracker H-M, Kreis H. When the crowd evaluates soccer players' market values: accuracy and evaluation attributes of an online community. Sport Manag. Rev. 2014 Nov;17(4):484-492.

[13] Hofmann J, Schnittka O, Johnen M, Kottemann P. Talent or popularity: what drives market value and brand image for human brands? J. Bus. Res. 2019 Jan;124:748-758

2) There is a lack of measures of goodness-to-fit of models used.

Following this comment, measures of goodness-to-fit have been incorporated in Table 4 on page 23 and 24. Additionally, the following text has been added:

“Table 4 shows the goodness of fit calculated using the RMSE, mean absolute error (MAE), and R2 on the training set.”. 

“According to Table 4, model 3 is the model that best fits the data for all three methods, with the GBM being the method with the lowest RMSE (9,442,425) and MAE (5,409,197), and the highest R2 (0.797)”. 

3) Joining in one group of footballers playing on different positions on the pitch is not correct (different roles on the pitch) - this problem [look work of Majewski, 2016] influences the popularity of the player (goalkeepers or defenders have not to be on the top of best scorers list - other features are significant for the goalkeeper and other for the striker. The such limitation should be taken into account, because different features may determine the popularity of players depending on the game's position.

We appreciate your remark regarding the inclusion of football players from different positions in the same group, and we agree that each position has distinct roles and features that determine a player's popularity.

In our approach, we pooled all players and modeled the effect of the the position with a categorical variable, following what was done by previous researchers [4, 5, 8, 10].

Furthermore, we included this concern as future research in the conclusions:

‘‘Future research could focus on creating position-specific models in the manner of Majewski [19], since certain game actions, such as goals or saves, may have greater significance for forwards or goalkeepers, respectively.’’ - Page 30

References:

[4] Frick B. The football players' labor market: empirical evidence from the major European leagues. Scott. J. Political Econ. 2007 Jul;54(3):422-446.

[5] Müller O, Simons A, Weinmann M. Beyond crowd judgments: data-driven estimation of market value in association football. Eur. J. Oper. Res. 2017 Dec;263(2):611-624.

[8] Singh P, Lamba PS. Influence of crowdsourcing, popularity and previous year statistics in market value estimation of football players. J. Discrete Math. Sci. Cryptogr. 2019 Feb;22(2):113-126.

[10] Garcia-del Barrio P, Pujol F. Hidden monopsony rents in winner-take-all

Markets-Sport and economic contribution of Spanish soccer players. Comput.

Stat. Data Anal. 2007 Jan;28(1):57-70.

4) The authors made their research on 1428 players from BIG-5 leagues from the years 2018-2019. There is no information about the type of time series used (daily, weekly or monthly). 

Thank you for your comment and for bringing up the issue of clarity on the type of time series used in our research. Further clarification on this point has been added on pages 17 and 19:

“The information from the players’ weekly popularity time series […] popularity information for each week of the players”. Page – 17.

“Consequently, 54 values (popularity per week) were stored for each player and used to construct the popularity indicators”. Page - 19

Regarding the variables related to player performance, they were measured as the average number of actions per game, calculated over the entire season. 

“Regarding the variables related to player performance, the game actions are measured as the average number of actions per game, calculated at the end of the season.’’ - Page 18.

Structurally, the five leagues used in the research differ from each other. So, there could exist also heterogeneity in the field of leagues. In my opinion, the results obtained for such heterogeneous objects are incorrect. Please, explain this doubt. Maybe there could be used some dummy variables representing different leagues? Maybe it could be done also for years.

We agree that there may be structural differences among the leagues that could affect the results obtained. The suggestion to include categorical variable leagues has been followed. However, it should be noted that only one season was used and that the explanatory variables are calculated as the average number of actions per match over the entire season. Therefore, it was not necessary to include some dummy variables for years. Additionally, according to referee 2's comments, we also introduced another categorical variable for indicating players’ nationality (specifically the continent).

“Finally, the league and continent variables were added to the models as categorical variables. Thus, players were classified according to the league in which their home team competed (LaLiga, Premier League, Serie A, Bundesliga, and Ligue 1). The second categorical variable represented the continent of birth of each player (Europe, Asia, Australia, Africa, North America, and South America).” - Page 19

5) Page 11/26 - the predictive analysis should be run after estimation and verification - there is a lack of verification of models.

Thanks for bringing this up. We agree that this point was not clear in our original manuscript. We have now revised the paper to improve clarity. To this end, Data section has been revised to include the creation of the training and test sets, and the description of the validation process of the models (see page 23):

“Then, the training set is created with k-1 subgroups, and the validation set (to validate the model) is made with the remaining subgroup. This process validates the model on non-overlapping datasets in each iteration, and the process concludes after all individuals have been in the validation set once (and only once) [49]”.

In addition, further clarification on this point has been added in the Results section on page 23:

“Machine learning culture was applied to validate and compare the models. Following Breiman [50], repeated k-fold cross-validation was used to avoid overfitting, and then, with the final selected model, we predicted the market values of the test set.” – Page 23

[50] Breiman L. Statistical modeling: the two cultures. Stat. Sci. 2001 Aug;16(3):199-231.

6) There is a lack of results for RF analysis and the GBM model. What do models look like? Which variables are included? Which variables are statistically significant?

Thank you for your remark. Detailed information on RF and GBM models, including variable selection, is included in Fig 1 (page 26) and S1-S4 Figs (Supporting Information section).

In addition, as RF and GBM are ensemble methods, they calculate the importance (RF) or the contribution (GBM) of each variable to the prediction. The complete explanation of the measures of variable importance and contributions is included in sections Random Forest and Gradient boosting modelling (see pages 20, 21 and 22).

7) Authors run the discussion only with one research - authors show differences between their own and corresponding research. Is there any other research? 

Unfortunately, this point was not clearly presented in our original manuscript, and we apologize for it. We have since revised the paper to provide a more thorough explanation.

The discussion was focused on the comparison with [5] because, to the best of our knowledge, it was the only research work using Google Trends (GT) to measure popularity. One of the contributions of our paper is the development of a methodology to obtain GT indicators for popularity, which is why we primarily compare it to the aforementioned work.

Reference:

[5] Müller O, Simons A, Weinmann M. Beyond crowd judgments: data-driven estimation of market value in association football. Eur. J. Oper. Res. 2017 Dec;263(2):611-624.

Following your comment and that of Referee 2, we have extended our discussion to include more studies (see pages 27 and 28).

“In the case of the variables related to player characteristics, the three models are in line with the results of previous research [5, 7, 11, 17, 18], which found that Age is a variable with a high contribution to the market value, whereas H and FT do not make essential contributions. In the case of the P variable, our results differ from those of previous studies [4, 5, 8], which found that this variable was statistically significant. None of our models found this variable to be important or statistically significant in predicting the players' market value. As for the continent variable, which stores information on the nationalities of the players, the methods used to carry out the prediction differ. In the case of MLR, the results showed that players born in South America are overvalued, as the coefficient was positive and statistically significant (see Table 6). This result is in line with the work of Pedace [16]. However, neither the RF method nor the GBM method found this variable to be important or statistically significant in predicting players' market value (see Fig 1 and S2 Fig).

In terms of the variables related to player performance, this study agrees with previous research since it highlights the importance (Fig 1 and S2 Fig) and statistical significance (Table 6) of variables such as PT [5, 8, 15], Gls [5, 8, 11], Dr [5, 7, 8], and S [7]. Note that S was not statistically significant in any model fitted by the MLR method. It is highlighted that although the variable DP has not been analysed in previous research, it was found to be either important (Fig 1 and S2 Fig) or statistically significant (Table 6) for all three methods. As far as the labour and club characteristics of the players are concerned, the results for the Ct variable (see Table 6, Fig 1, and S2 Fig) are largely in line with those of previous researchers [15, 21, 24], who pointed out that the contract's duration influences the players' market value and in fact increases this value. Finally, in terms of players' leagues, the MLR method indicates that players competing in Ligue 1 and Serie A are undervalued. In contrast, the opposite is true for players in the Premier League (Table 6). This finding agrees with the results of previous researchers, who found statistically significant differences between players competing in the Premier League and players competing in other leagues [17, 18]. However, in the case of the RF and GBM methods, even though in model 2 (S1-S3 Figs), competing in the Premier League contributes to predicting the market value of the players, this effect is not found in model 3 (Fig 1 and S2 Fig).”

Finally, note that according to [3]: 

“Popularity (variables) are more rarely included in analysed articles.”

Therefore, this research gap prevents carrying out a more in-depth comparative study.

[3] Franceschi M, Brocard JF, Follert F, Gouguet JJ. Determinants of football

players' valuation: a systematic review. J. Econ. Surv. 2023 Feb; 1-24. 

Response to referee 2:

We appreciate the changes and suggestions indicated and your detailed reading of the paper. The changes made are detailed below.

o Abstract

1) The abstract should be more straight forward and could need a bit of restructuring to make it clearer and to have a better understanding at first glimpse. What is the research question, what are the methods used, what are the results and findings, as well as the implications? 

Thanks to your comment, the abstract was clarified. 

“Google Trends is a valuable tool for measuring popularity since it collects a large amount of information related to Google searches. However, Google Trends has been underused and misused by sports analysts. This research proposes a novel method to calculate several popularity indicators for predicting players' market value. Google Trends was used to calculate six popularity indicators by requesting information about two football players simultaneously and creating popularity layers to compare players of unequal popularity. In addition, as the main idea is to obtain the popularity indicators of all players on the same scale, a cumulative conversion factor was used to rescale these indicators. The results show that the proposed popularity indicators are essential to predicting a player's market value. In addition, using the proposed popularity indicators decreases the transfer fee prediction error for three different models that are fitted to the data using the multiple linear regression, random forest, and gradient boosting machine methods. The popularity indicator Min, which is a robust reflection of the popularity that represents a player's popularity during the periods when they are less popular, is the most important popularity indicator, with a significant effect on the market value. This research provides practical guidance for developing and incorporating the proposed indicators, which could be applied in sports analytics and in any study in which popularity is relevant.”

o Introduction

2) You write that “The sports industry is a scenario where most interests converge, from political to public, increased by communication media and information.” I am not sure I understand this sentence. Could you reformulate it please.

We genuinely appreciate this comment, and the sentence was clarified:

‘The sports industry is a place where most interests, from political interests to public interests, converge, and traditional media and social media have helped to increase football's popularity worldwide’’ - Page 2

3) What exactly do the authors refer to when mentioning “income”? Club revenues, Gross income, net income? Player income?

We are referring to club revenues. The sentence has been revised to further clarify the point: 

‘‘Indeed, the popularity of this sport has caused football clubs to generate increasing amounts of revenue.’’ - Page 2

4) You state “Football club revenues are often closely related to teams' investment in their main assets: the players”, as has been previously pinpointed by Frick (2007). But what do you exactly mean with this sentence, that clubs with higher revenues are able to purchase players that have higher transfer fees?

Following your comment, the sentence rephrased for clarity:

‘’Football players have already been recognised as accounting assets for clubs [2], and their valuation is an essential indicator of the financial value of teams [3]. Thus, from both sports and business points of view, the revenues of football teams are often closely related to the teams' investment in their main assets: the players.’’ - Page 2

5) Since you study a probable effect of “popularity” on “market value”, you should first of all properly assess the two key variables, i.e. popularity and market value. Popularity is quite detailed described in section 1.2. As of “market value” you cite Herm et al. (2014) with “an estimate of the amount of money that a club would be willing to pay for an athlete contract, regardless of an actual transaction”. However, this definition has been criticized in the past. What “club” do Herm et al. refer to? Does the term “market” not rather suggest that we are not considering one club but several clubs or several stakeholders? How do we know what a single club is willing to pay if they do not communicate it? And in negotiation process or even after, it is highly unlikely that a club would communicate the prize they would have been willing to pay. Given that in your study you do not ask one or several clubs for their willingness, but rather obtain your market values from transfermarkt, you should be consistent here. A discussion of transfer and market value has been provided in a recent working paper by Franceschi et el. (What can football economics literature learn from subjective value theory?). They also provide an overview of alternative definitions in their appendix. 

Thank you for this very precise comment. We have improved the discussion and updated the market value definition. Specifically, we have used the Transfermarkt definition, as [3] did. 

[3] Franceschi M, Brocard JF, Follert F, Gouguet JJ. Determinants of football players' valuation: a systematic review. J. Econ. Surv. 2023 Feb; 1-24.

“According to Franceschi et al. [3], researchers have recognised the empirical proximity between players' transfer fees and their market value. Therefore, these values are comparable [4] since they store similar information and are influenced by the same variables. The transfer fee is ‘the actual price paid on the market’ [5] by a football team for a player at a given time, and it is rarely available to the public [3]. Thus, to solve the problem of the lack of information on the actual transfer fees, researchers and fans have started paying attention to websites that offer estimates of the market value of players. An example is Transfermarkt, which, although the website itself explains that its goal is not to predict player transfer fees but to provide the ‘expected value of a player in a free market’ [6], has gained great prestige not only among football industry professionals (coaches and journalists) but also among scientists who have found this website helpful for estimating the market value [5, 7, 8].”- Page 3.

o Background

6) The section is rather short and the summary of previous research performed in the three described parts (Player characteristics, Player performance, and Player popularity) is not elaborately enough and too unspecific. I suggest you explain the findings in much more detail and the partly conflicting results of different existing studies. The results are not as unitary as you make it sound. Please have a look at Franceschi et el. (2023) for a recent literature review on football player valuations.

A more thorough revision of previous research has been performed, and a more elaborate background has been carried out. Specifically, we have expanded on the Player characteristics, and Player performance sections (Page 6 to 10).

“Player characteristics

Previous research has considered player characteristics, e.g. a player's age, height, footedness, nationality, and position and contract (years until the end of the contract); these quantities represent physical or market information, as they are variables related to the market value [5, 7, 9, 15, 16]. 

The variable age (age2 is also used) is a variable that most studies include and generally find statistically significant [5, 7, 11]. Müller et al. [5], who used age2, found that this variable was statistically significant, with a negative effect on the players' market value. Herm et al. [11] found the same effect, although they used an age-corrected linear function. Behravan and Razavi [7], who considered how independent variables affect the player valuation for different positions, found that age influenced the market value for all positions except goalkeepers. In addition, some studies have even used the quarter of the year in which each player was born as an explanatory variable and found evidence that it is related to the market value [17, 18]. According to Felipe et al. [17], the most valued players are attacking midfielders born during the first quarter. Gyimesi and Kehl [18] studied the effect of the relative age on the market value, concluding that it is higher at younger ages and decreases as the player ages. However, some research found a non statistically significant relationship between age [10, 19] or age2 [10] and the market value. Finally, Franceschi et al. [3] considered that the importance of the age variable and its positive impact (negative when the quadratic form is used) on the market value of players is due to the relationship between the experience and age of the players, i.e. the older the player, the more experience they have, but the older the player, the lower the player's physical potential.

Most researchers agree that the position is an influential variable for the players' market value [4, 5, 8, 10]. Garcia-del Barrio and Pujol [10] used the position to control for heterogeneity associated with a player's position on the field. Their results showed that forwards have a higher valuation than goalkeepers (the reference used in their regression models). Frick [4] determined that the position significantly influences player salaries, with goalkeepers being the players with the lowest salaries. Müller et al. [5] considered four multilevel models, which included the position as a random effect, to capture the dependence between the market values of the players who played in the same position. Other authors, such as Behravan and Razavi [7] and Majewski [19], estimated the market value of players by grouping them according to their positions and predicting the market value for each group of players independently. In addition, the explanatory variables used to predict the players' market value were selected depending on their position on the field (goalkeeper, defender, midfielder, or forward) [7, 19].

Furthermore, Herm et al. [11] and Bryson et al. [20] argued that the ability to ‘two-foot’ could improve a player's market value, as it allows the player to play in different positions. Behravan and Razavi [7] also considered it necessary for forwards to be able to use both feet; in particular, they reflected on the importance of being able to use the ‘weak foot’ accurately. However, Garcia-del Barrio and Pujol [10] and Müller et al. [5] did not find that the footedness variable was statistically significant. Finally, previous analyses have examined the effect of nationality on the market value [5, 10, 16]. Garcia del Barrio and Pujol [10] found that European players tended to be overvalued in LaLiga (Spanish first division), while non-European players were undervalued. Pedace [16], who analysed discrimination based on nationality in the Premier League (English first division), found evidence that South American players were overvalued. Müller et al. [5] also included the continent of origin as a random effect to capture the dependence between the market values of players born on the same continent.”

“Player performance

The player performance variables measure players' actions during the season. The playing time is a standard statistically significant variable researchers use to determine the market value [5, 8, 15]. In addition, studies have found a positive and significant relationship between the number of goals and assists and the market value [5, 8, 11, 19]. The consistency and importance of both variables have led some researchers to develop new study methods. For example, Herm et al. [11] developed the scoring variable, which is calculated as the number of goals and assists corrected by the average for the player's position. According to their results, the higher the number of goals or assists scored, the higher the market value. Majewski [19] developed a synthetic variable using the sum of the goals and assists, obtaining a positive and statistically significant result.

Behravan and Razavi [7] developed the shooting skill variable, which stored information on accuracy, instead of studying the relationship between the number of goals and the market value. The results indicated that the shooting skill was essential to the market value for defenders and forwards.

Regarding the defensive variables, the number of interceptions, yellow and red cards, and fouls have also been occasionally used [5, 19]. Majewski [19], who introduced yellow and red cards into the model, found that neither variable was statistically significant. According to Müller et al. [5], only the number of yellow cards was statistically significant. According to Franceschi et al. [3], although the number of yellow cards and the number of red cards are easy and inexpensive variables to calculate, their use may not be recommended due to their repeated lack of significance.

Even though these variables are used more infrequently, previous research has considered the passing accuracy [5, 7, 11], the number of dribbles [5, 7], and the number of duels [5, 11]. Herm et al. [11] defined the variable accuracy as the percentage of successful passes corrected for the average for the player's position. Based on their results, the more accurate the player's passes are, the higher their market value will be. Similarly, Herm et al. [11] calculated the number of won duels, or the save-to-shots ratio for the goalkeeper, corrected for the average for each position, and found that it had a significant effect on the players' market value. Specifically, Müller et al. [5] considered the following explanatory variables of the market value: the number of passes, successful passes, number of aerial duels, successful aerial duels, number of dribbles, and successful dribbles. However, only the number of passes, successful passes, the number of aerial duels, and the number of dribbles were positive and statistically significant. Behravan and Razavi [7] studied the effect of the passing and dribbling abilities of the players according to their positions. They found that dribbling should be considered when the relationship of the variables that affect the market value of forwards is studied. Furthermore, their results showed that long passes (in the case of defenders and midfielders) and the passing ability (in the case of forwards) were essential factors in estimating the players' market value.”

7) You write that “Researchers have identified three principal groups of variables for analysing a player's market value: Player characteristics, Player performance, and Player popularity.” Since you site Frick (2007) you are aware that there are a few more factors. You could further add a section for other influencing factors that cannot be summarized under these three categories but that have been found to influence market value in the past, such as current market conditions, current and future clubs, leagues of the selling and purchasing club, the nationality of a player – in fact, some of these factors could be factors that influence market value via your variable popularity. A player that has been playing for a relatively short time at a club outside Europe and is suddenly purchased by a top club receives a big boost in terms of popularity (latest example Endrick from Palmeiras who will be transferred to Real Madrid in 2024)

Following your suggestion, a new section has been added to account for other variables that may also influence the market value of players. In addition, two variables have been added to our predictive models: players’ league and nationality (specifically the continent). See subsubsection Labour and club characteristics of the players (Pages 10 and 11).

“Labour and club characteristics of the players

The group of variables related to the labour and club characteristics of the players includes variables related to a player's employment contract (e.g. the length of the contract), the league in which the player competes, previous player valuations, and information that is not directly related to the player's performance but is related to their club's performance (e.g. the FIFA ranking of the teams).

Previous analyses have examined the effect of the contract length on players' market value and found evidence that the longer the contract expiration date, the higher the transfer fee [15, 21, 24]. Furthermore, Franceschi et al. [3] argued that, given that at the end of a player's contract, their transfer value is zero, if the time remaining until the end of the contract is not included in market value prediction models, relevant information affecting the transfer fee of players could be lost.

Other valuations, such as the previous market value, have also been used to predict the market value [5, 8]. To estimate the player's market value after a given season, Müller et al. [5] used, as an explanatory variable, the log transformation of the Transfermarkt estimate of the player's market value from the end of the previous season. Similarly, Singh and Lamba [8] used the previous market value estimated by Transfermarkt to predict the current market value of players. In both investigations, the variables were positive and statistically significant.

The league in which the player competes is a categorical variable introduced as a random effect to capture the non-independence between players' market values and the league in which they compete [5, 24]. Felipe et al. [17], who analysed the interaction of different explanatory variables with the Transfermarkt market value, also took the league into account. According to their findings, there were differences between the market values of players competing in the Premier League and the market values of players in other leagues. Gyimesi and Kehl [18] obtained similar results and found statistically significant differences in the market values of similar players among the Big Five.

Furthermore, previous researchers considered the impact of players' presence in European championships and on their national teams in predicting their market value [8, 10]. Garcia-del Barrio and Pujol [10] counted how many matches a player had played in Europe (considering both the Champions League and the UEFA Cup) and how many matches the player had played for their national team. For both variables, quadratic terms were also considered. Garcia-del Barrio and Pujol [10] found that players who had competed in Europe and with their national teams had a significantly higher market value. Singh and Lamba [8] used the valuations of players provided by the FIFA 2018 game as an explanatory variable and found that this value impacted the players' market value.

Finally, how a local or national team's reputation influences players' market value has also been analysed [7, 19]. Majewski [19] considered the FIFA national team ranking as an integer from 1 (the best national team) to 209 (the worst national team). According to their results, since the coefficient of this variable was negative and statistically significant, a lower ranking of the national team decreased the value of the players. According to the study carried out by Behravan and Razavi [7], the reputation of the national team, measured based on the FIFA world ranking, is important for estimating defenders' and midfielders' market value.

8) In the section on Player characteristics, you refer to previous analyses that have found contract length to influence performance. What do you want to tell the reader here? Is remaining contract length a player characteristic? And why do you write about performance as an endogenous variable here?

Following this comment, the length of the contract was moved to the new section. Please see subsubsection Labour and club characteristics of the players (Page 10 and 11). In addition, we have clarified how the contract length variable has been used in previous research.

“The group of variables related to the labour and club characteristics of the players includes variables related to a player's employment contract (e.g. the length of the contract), the league in which the player competes, previous player valuations, and information that is not directly related to the player's performance but is related to their club's performance (e.g. the FIFA ranking of the teams).

Previous analyses have examined the effect of the contract length on players' market value and found evidence that the longer the contract expiration date, the higher the transfer fee [15, 21, 24]. Furthermore, Franceschi et al. [3] argued that, given that at the end of a player's contract, their transfer value is zero, if the time remaining until the end of the contract is not included in market value prediction models, relevant information affecting the transfer fee of players could be lost.”

9) You write that “while the characteristics and performance of players have helped predict their transfer fees, they are not perfect measures of a football player's market value, so complementing them with popularity data could offer a better idea of a player's valuation.” This assertion falls somewhat from the sky and you do not refer to previous research nor have a comprehensible rationale for this claim. Please elaborate.

Thank you for your comment. We have revised the text to improve clarity and provide the reader with a more elaborate background of the assertion made:

“While the characteristics and performance of players have helped predict their transfer fees, they are not perfect measures of a football player's market value; researchers such as Franceschi et al. [3] have highlighted that variables such as contracts, youth academies, and popularity have strong theoretical justifications and promising empirical results that indicate that they can better explain the transfer market. In the same way, previous researchers have pointed out that popularity-related data are an interesting complement that can improve a player's valuation [5, 10, 11, 13]. Garcia-del Barrio and Pujol [10], who incorporated popularity ratings into their study, expressed their concern that (before then) no study related to the sports industry had included popularity variables, as these turned out to have an important role in helping to explain economic behaviour in the sports industry. Herm et al. [11] emphasised that it is necessary to study variables related to popularity, as these variables impact the market value of athletes. Hofmann et al. [13] highlighted that managers are interested in knowing whether performance or popularity is more important in boosting the players' market value. Müller et al. [5] concluded that not only player characteristics and performance but also measures of player popularity could help football teams make predictions concerning the evolution of the market value.” - Page 12

10) You write that “…the brand image is also sold or exchanged with the player…it is necessary to study the specific effect of popularity on the transfer fee.” The concept of image rights, as understood today, stems from the so-called “right of publicity” (Al-Ameen, 2017). However, the ownership and control of image rights, which can be exploited through sponsorship, endorsement and merchandising, is not subject to legal rules on allocation of rights and responsibilities but is a mere question of negotiations between clubs and players (Haynes, 2007). It is common practice that clubs enter into employment contracts with their athletes that require them to cede the use of their brand or image for marketing and commercial activities (Hewison, 2001). While a few powerful football club such as Arsenal, Manchester United and Real Madrid demand full control over players’ image rights, other clubs are just content to get a minimal share of the proceeds from the commercial exploitation of the player’s image rights (Al-Ameen, 2017). Star players in top leagues strive to circumvent the image right obligations by, often successfully, leveraging their position to obtain advantageous conditions, whereas average players are often denied this privilege.

Thank you again for this deep reflection. Based on it, we include the following argumentation in the text:

“Hofmann et al. [13] argued that most clubs own the branding rights of their players, which brings additional revenue to the team. However, the concept of image rights (or branding rights), understood today as ‘publicity rights’ [27], is not subject to legal rules but depends on contractual negotiations between players, agents, and teams [28]. As Hofmann et al. [13] indicated, it is typical for a football club to own the publicity rights of the players, who, in most cases, are required to cede the use of trademarks or sponsorships for marketing and commercial activities that provide financial profitability to the team [29]. However, not all clubs exert the same pressure on the control of publicity rights; while the most powerful teams (e.g. Arsenal, Manchester United, and Real Madrid) demand complete control, others agree to obtain a minor portion of profits from the commercial exploitation of players' publicity rights [27]. Thus, while ‘superstars’ strive to maintain complete control over their image rights, average football players are often denied this privilege. Therefore, given the impact of a transfer on the economy of the clubs, it seems necessary to study the specific effect of popularity on the transfer fee.” - Page 13

11) Some rather recent articles using machine learning technics in determining market value have not been considered at all, such as 

Thank you for bringing this to our attention. We included the suggested references in the revised version of the manuscript.

“In addition, previous researchers have used these machine learning techniques to determine the market value using the RF [8] and gradient boosting, namely XGBoost [36-38].” – Page 19

[8] Singh P, Lamba PS. Influence of crowdsourcing, popularity and previous year statistics in market value estimation of football players. J. Discrete Math. Sci. Cryptogr. 2019 Feb;22(2):113-126.

[36] McHale IG, Holmes B. Estimating transfer fees of professional footballers using advanced performance metrics and machine learning. Eur. J. Oper. Res. 2023 Apr;306(1):389-399.

[37] Yigit AT, Samak B, Kaya T. An XGBoost-lasso ensemble modeling approach to

football player value assessment. J. Intell. Fuzzy Syst. 2020 Jan;39(5):6303-6314.

[38] Zhang D, Kang C. Players' value prediction based on machine learning method.

In: Proceedings of the Journal of Physics: Conference Series, Volume 1865. 2021

International Conference on Advances in Optics and Computational Sciences

(ICAOCS); 2021 Jan 21{23; Ottawa, Canada. IOP Publishing; 2021. p. 042016.

o Material and methods

12) What was the reason you only included players of the big 5 leagues? Maybe you could perform the analysis by additionally using players of other popular leagues such as the Eredivisie, German Bundesliga 2, Portuguese Primeira Liga, and Belgium First Division where there is enough information available and the players are known, thus do not cause problems with market value. Additionally, these are the leagues where a lot of players of the Big 5 leagues are bought from. This way you would have a bigger dataset than that of 1428 players, which is a rather small dataset (as you pointed out yourself in your concluding remarks) – sample size is important, especially in a machine learning context and the degrees of freedoms you have.

Although including other popular leagues out of the Big Five is really interesting, we use the Big Five because, in terms of comparison is richer [8, 17, 18, 20]. In addition, [5] highlights a Transfermatk limitation on page 612 subsection 2.2: 

“Fourth, authors who have analysed the effect of popularity have not studied how much of the transfer fee is explained by it.”

Moreover, we agree that including more players could increase the predictive power of the model. However, the paper's main objective is to evaluate how popularity influences the players' valuation and whether this can be measured using Google Trends. Therefore, expanding the player base could lead to problems like the ones described by [5] and could divert us from the objective of measuring popularity.

References:

[5] Müller O, Simons A, Weinmann M. Beyond crowd judgments: data-driven estimation of market value in association football. Eur. J. Oper. Res. 2017 Dec;263(2):611-624.

[17] Felipe JL, Fernandez-Luna A, Burillo P, de la Riva LE, Sanchez-Sanchez J, Garcia Unanue J. Money talks: team variables and player positions that most influence the market value of professional male footballers in Europe. Sustainability. 2020 May;12(9):3709.

[18] Gyimesi A, Kehl D. Relative age effect on the market value of elite European football players: a balanced sample approach. Eur. Sport Manag. Q. 2021 Mar;0(0):1-17.

[20] Bryson A, Frick B, Simmons R. The returns to scarce talent: footedness and

player remuneration in European soccer. J. Sports Econ. 2013 Dec;14(6):606-628.

13) Do you have examples of other studies where the model has been trained with one data set (in your case market values) and then tested with another dataset (transfer fees)?

Yes, we revised the text to indicated this (page 17). Specifically, that procedure was carried out by [5] (on page 618 subsection 5.3). 

“Therefore, the models were trained using the market values estimated by Transfermarkt as the dependent variable, although the model error was computed using the transfer fees of the players that were signed. Müller et al. [5] previously carried out this procedure.”

14) As for the dependent and explanatory variables used, there might be the need to add a variable indicating player injuries. Injuries are very common in the game of football, and while short to midterm injuries have little impact on market values, they certainly impact virtually all player performance variables.

This comment is really interesting. We agree that injuries can influence market value. However, an extensive review of the determinants of player valuation did not report any research considering this variable [3], possibly because of the lack of data. Therefore, we have included this limitation in the conclusions:

“Finally, another interesting variable that future research should consider is player injuries. Although this information has not been widely used so far, it can influence the players' performance variables, especially if the injury occurs at the beginning of the season. Therefore, it could also impact the players' market value.’’ - Page 31

[3] Franceschi M, Brocard JF, Follert F, Gouguet JJ. Determinants of football players' valuation: a systematic review. J. Econ. Surv. 2023 Feb; 1-24

15) Previous research has found evidence for customer and employer-based discrimination based on nationality in the case of football (Pedace, 2008). Players from highly populated and football enthusiastic countries such as Mexico and Brazil might have distorted popularity figures when comparing these figures to their market values. One such case was James Rodriguez when playing for Bayern Munich between 2017 and 2019. In your study you do not control for nationality or home population

The suggestion of including some dummies representing the nationality has been followed. Specifically, following previous researchers, we classify the variable nationality according to the continent [5]. According to referee 1's comments, we also introduced another categorical variable for indicating the leagues.

“Finally, the league and continent variables were added to the models as categorical variables. Thus, players were classified according to the league in which their home team competed (LaLiga, Premier League, Serie A, Bundesliga, and Ligue 1). The second categorical variable represented the continent of birth of each player (Europe, Asia, Australia, Africa, North America, and South America).” - Page 19

16) You used 5 different leagues and the market values as provided by transfermarkt with two different currencies, EUR and GBP. Are you aware that the transfermarkt data calculates past transactions at current exchange rate? How do you account for these exchange rate discrepancies? 

We appreciate the reviewer’s comment concerning the exchange rate discrepancies. We would like to clarify that the Transfermarkt market value was taken at the same time as the rest of the variables in the summer of 2019 (because this database was used for my Master’s thesis). Since all of the obtained data was transformed to euros in summer 2019, any subsequent exchange rate changes have no impact on our work. However, this is a problem we would face when enlarging the base.

17) You might want to explain why you have chosen random forest and gradient boosting machine among the many machine learning models that are used as of today.

Thank you for your comment; we have added to the text (page 19) why we have selected these methods. 

“The multivariate methods selected to train and test the models were the multiple linear regression (MLR) [32], random forest (RF) [33], and gradient boosting machine (GBM) [34, 35] methods. MLR is a classical method that is used as a benchmark, and the RF and GBM methods were selected due to the good results they have achieved in most of the research fields in which they have been applied. In addition, previous researchers have used these machine learning techniques to determine the market value using the RF [8] and gradient boosting, namely XGBoost [36-38]. Note that both RF and GBM make it possible to analyse the importance of a variable.”- Pages 19 and 20

Note that although other methods, such as Support vector machines or neural networks, could provide sometimes more accurate results, the fact that those models are black boxes does not allow us to know whether the constructed popularity indicators are important for predicting the market value of players, whereas RF and GBM allow us to do so.

18) Table 4 needs a makeover, it is hard to read and interpret. Having said that, the results of the MLR are not discussed. 

In response to the reviewer’s comment, Table 4 (now Table 5) was better organized for comparison purposes and considering the journal’s requirements. Page 24.

19) There is only a comparison between the models.

Following this comment, we have extended the analysis of the selection of variables to the rest of the methods (Please see pages 26 and 28).

“Fig 1 shows that the most crucial variable for predicting the market value of a player signed during the summer market of the 2018/2019 season is the popularity indicator Min, which contains information about the week in which the player was searched the least. Note that Min was the most important variable in the case of the GBM (Fig 1) and RF (S2 Fig) methods, and it was statistically significant in the case of MLR (Table 6).”

“In the case of the variables related to player characteristics, the three models are in line with the results of previous research [5, 7, 11, 17, 18], which found that Age is a variable with a high contribution to the market value, whereas H and FT do not make essential contributions. In the case of the P variable, our results differ from those of previous studies [4, 5, 8], which found that this variable was statistically significant. None of our models found this variable to be important or statistically significant in predicting the players' market value. As for the continent variable, which stores information on the nationalities of the players, the methods used to carry out the prediction differ. In the case of MLR, the results showed that players born in South America are overvalued, as the coefficient was positive and statistically significant (see Table 6). This result is in line with the work of Pedace [16]. However, neither the RF method nor the GBM method found this variable to be important or statistically significant in predicting players' market value (see Fig 1 and S2 Fig).

In terms of the variables related to player performance, this study agrees with previous research since it highlights the importance (Fig 1 and S2 Fig) and statistical significance (Table 6) of variables such as PT [5, 8, 15], Gls [5, 8, 11], Dr [5, 7, 8], and S [7]. Note that S was not statistically significant in any model fitted by the MLR method. It is highlighted that although the variable DP has not been analysed in previous research, it was found to be either important (Fig 1 and S2 Fig) or statistically significant (Table 6) for all three methods. 

As far as the labour and club characteristics of the players are concerned, the results for the Ct variable (see Table 6, Fig 1, and S2 Fig) are largely in line with those of previous researchers [15, 21, 24], who pointed out that the contract's duration influences the players' market value and in fact increases this value. Finally, in terms of players' leagues, the MLR method indicates that players competing in Ligue 1 and Serie A are undervalued. In contrast, the opposite is true for players in the Premier League (Table 6). This finding agrees with the results of previous researchers, who found statistically significant differences between players competing in the Premier League and players competing in other leagues [17, 18]. However, in the case of the RF and GBM methods, even though in model 2 (S1-S3 Figs), competing in the Premier League contributes to predicting the market value of the players, this effect is not found in model 3 (Fig 1 and S2 Fig).”

o Discusion

20) How do the results of the paper contrast to other studies that examined the impact of player popularity on transfer fees? 

Concerning results related to popularity indicators, as far as we know, Google Trends (GT) has been used only by [5]. Following your comment, the revised paper results are contrasted with previous research works about popularity:

“Although the effect of players' popularity on the market value has been explored [5, 8, 10, 11, 13], this study proposes new ways to use GT data to measure players' popularity by calculating PIs. GT is a valuable tool that contains information on Google searches by name, in the image, news, and shopping categories, or on YouTube. Therefore, while previous researchers have used data from different web sources (e.g. the number of visits to Wikipedia, YouTube views, or Google visits), GT summarises all this information in a single value that provides a global view of the popularity of the players. In addition, while most of the variables listed above store information at a specific point in time, GT provides information over a time range. Another problem that GT avoids is missing data because, unlike variables such as the number of followers on Instagram, Facebook, or Twitter that depend on players having social media accounts, GT considers any appearance on Google. Finally, with GT, we can mitigate the problem of multicollinearity, as all popularity information is stored in a single variable.” - Page 28

In addition, the conclusions were also extended:

“The results obtained in this study agree with the results of previous investigations, which indicated that the variables related to the players' popularity contribute to predicting the players' market value [5, 10, 11, 13].” - Page 29

21) Please detail the implications of this paper and your results for management

We thank the referee for this suggestion, so we have added the implications of our results in terms of management:

“The results emphasise the importance of players' systematic brand images on the market value and, therefore, the revenue of clubs. Therefore, clubs and agents should focus on maintaining or improving the positions of the players and establishing mechanisms to increase their popularity. However, it should not be forgotten that performance-based attributes have direct and indirect effects on player popularity, and not the other way around [13]. Thus, there are two key conclusions: a) players have to enhance their performance attributes, which will support their media-related attributes, and b) managers and clubs have to apply successful strategies to improve or maintain the brand images of players.” - Page 30

---

## [Decision Letter · Decision Letter 1]

19 Jun 2023

PONE-D-22-31349R1

Measuring the popularity of football players with Google Trends

PLOS ONE

Dear Dr. Malagón-Selma,

Thank you for submitting your manuscript to PLOS ONE. After careful consideration, we feel that it has merit but does not fully meet PLOS ONE’s publication criteria as it currently stands. Therefore, we invite you to submit a revised version of the manuscript that addresses the points raised during the review process.

Reviewer 2 addresses some points that I agree with after my own reading. In particular, the more critical presentation of the so-called "market values" from the transfermarkt.de platform is essential. A broad literature uses these largely unreflectively as dependent variables in their models. I would like to ask you to take up the points of criticism. I assume that we can accept the paper after proper implementation without further rounds.

We look forward to receiving your revised manuscript.

Kind regards,

Prof. Dr. Florian Follert

Academic Editor

PLOS ONE

Reviewers' comments:

Reviewer's Responses to Questions

**Comments to the Author**

1. If the authors have adequately addressed your comments raised in a previous round of review and you feel that this manuscript is now acceptable for publication, you may indicate that here to bypass the “Comments to the Author” section, enter your conflict of interest statement in the “Confidential to Editor” section, and submit your "Accept" recommendation.

Reviewer #1: All comments have been addressed

Reviewer #2: All comments have been addressed

2. Is the manuscript technically sound, and do the data support the conclusions?

Reviewer #1: Yes

Reviewer #2: Yes

3. Has the statistical analysis been performed appropriately and rigorously? 

Reviewer #1: Yes

Reviewer #2: Yes

4. Have the authors made all data underlying the findings in their manuscript fully available?

Reviewer #1: Yes

Reviewer #2: Yes

5. Is the manuscript presented in an intelligible fashion and written in standard English?

Reviewer #1: Yes

Reviewer #2: Yes

6. Review Comments to the Author

Reviewer #1: (No Response)

Reviewer #2: First and foremost, I would like to express my gratitude for the effort put into enhancing the manuscript. I appreciate your responsiveness to my initial comments and the progress made in the paper. However, I must candidly state that the paper still requires substantial improvements, and I have several critical questions and comments that the authors should carefully consider. The questions and comments are presented below.

1. The final paragraph of the Introduction ("The remainder of the paper...") could be omitted as the manuscript adheres to a standard structure.

2. It is worth noting that the use of market values has faced significant criticism from some academics. It would be advantageous to provide a more critical assessment and acknowledge the limitations associated with employing websites like www.transfermarkt.de. For instance, Müller et al. (2017) criticize the use of such platforms (https://www.sciencedirect.com/science/article/pii/S0377221717304332).

3. In their paper on the impact of Club revenues on transfer fees and salaries, Quansah et al. (2021) demonstrate that club revenues are a crucial variable in determining transfer fees and player market values, revealing the vulnerability of these factors during times of crises. Maybe it would be good to consider the implications of their findings for the relevance of utilizing Google Trends as a measure of popularity during periods of distress in your own research.

4. Furthermore, it is mentioned that Cristiano Ronaldo is the most followed person on Instagram. Lionel Messi, another popular but « aging » football figure, closely trails him in terms of followers. While there certainly had been a strong correlation between popularity and market value earlier in their careers, it appears that this relationship has become nonlinear. Although their popularity continues to grow, their market values have experienced significant declines.

Additionally, Messi's transfer from PSG to Inter Miami in the US, a relatively inferior league, has garnered substantial media attention and had a noteworthy impact on the player's popularity (and google trends), which may not be (or is certainly not) reflected in the player's continuously decreasing market value. While these are only two examples of anecdotal evidence, there are certainly more cases.

5. The paper includes several unsupported claims, such as "However, Google Trends has been underused and misused by sports analysts." It is essential to provide evidence or support for such claims rather than making assertions without a factual basis. Similarly, the statement "The sports industry is a place where most interests, from political interests to public interests, converge, and traditional media and social media have helped to increase football's popularity worldwide" remains unsupported, along with certain other claims in the document. There are instances where the reader fails to discern the source of the information presented.

6. In the introduction, it is stated that the "Big Five" European football leagues generated revenues exceeding €9.283 million in the 2018/2019 season. As we are currently in 2023 and the 2022/23 season has concluded, it is necessary to update this information.

7. Additionally, the data used in the analysis pertains to the 2018/19 season. It is important to clarify the reason for analyzing such outdated data. Was it due to convenience, as the initial research was conducted four years ago?

8. The statement "Thus, to solve the problem of the lack of information on the actual transfer fees, researchers and fans have started paying attention to websites that offer estimates of the market value of players" suggests that football fans have turned to websites like transfermarkt.de due to the scarcity of transfer fee information. It would be beneficial to provide evidence or support for this claim.

9. The information regarding Cristiano Ronaldo's 308 million followers on Instagram may also be outdated ? The last time I looked he had close to 600 million followers.

While the manuscript has shown huge improvement and is engaging to read with a coherent narrative, there are notable inconsistencies within the document (unsupported claims and missing citations), and the utilization of outdated data from 2018/19 is a major concern. To address these issues, the scholars are advised to rectify the inconsistencies, possibly by involving a professional proof-reader, and update the model with more recent data. Alternatively, conducting a robustness check using more up-to-date data would also be valuable. Once these points are addressed, I recommend accepting the manuscript. I extend my best wishes to the researchers.

7. PLOS authors have the option to publish the peer review history of their article (what does this mean?). If published, this will include your full peer review and any attached files.

Reviewer #1: No

Reviewer #2: No

---

## [Author Response · Author response to Decision Letter 1]

10 Jul 2023

Prof. Dr. Florian Follert

Academic Editor

PLOS ONE

Valencia, 10th July 2023

 Dear Prof. Follert,

 We enclose the revised version of our article Measuring the popularity of football players with Google Trends (ref. PONE-D-22-31349) in which we have incorporated the referees’ changes and suggestions. We would like to thank the Reviewer for their insightful comments that have allowed us to really improve the original version of the paper.

Next, the changes made in the paper following the referee and editor recommendations are detailed and explained. 

In addition, we also enclose written responses in which we give an answer to each of the points made in their reports.

Yours sincerely,

Pilar Malagón

 

Response to Editor:

We appreciate the changes and suggestions and your detailed reading of the paper. The changes made are detailed below.

1) The more critical presentation of the so-called "market values" from the transfermarkt.de platform is essential. A broad literature uses these largely unreflectively as dependent variables in their models. I would like to ask you to take up the points of criticism.

Following the recommendation of the editor, as well as the suggestion of referee 2, we have provided a more critical explanation of the Transfermarkt website. Page 5:

“Numerous studies have acknowledged Transfermarkt as a useful approximation to players’ market value. However, such a crowdsourcing approach has its limitations. In this respect, M ¨uller et al. [7] point out that the objectivity of the community members, on whose opinions the valuation is based, cannot be guaranteed. The process of aggregating user valuations lacks homogeneity, making it difficult to reproduce the results. Moreover, the estimation of market value tends to have higher variability for less popular players receiving less attention and valuations. More recently, Franceschi et al. [5] stress the subjectivity already mentioned by [7] and emphasize the conceptual difference between the transfer fee and the estimated market value provided by platforms like Transfermarkt. Nevertheless, researchers have recognized the empirical proximity between these measures [5, 7, 13, 23, 26, 27].” 

Response to referee 2:

We appreciate the changes and suggestions indicated and your detailed reading of the paper. The changes made are detailed below.

1) The final paragraph of the Introduction ("The remainder of the paper...") could be omitted as the manuscript adheres to a standard structure.

We genuinely appreciate this comment, and the paragraph was omitted. 

2) It is worth noting that the use of market values has faced significant criticism from some academics. It would be advantageous to provide a more critical assessment and acknowledge the limitations associated with employing websites like www.transfermarkt.de. For instance, Müller et al. (2017) criticize the use of such platforms (https://www.sciencedirect.com/science/article/pii/S0377221717304332).

Thank you for this precise comment. We have provided a more critical explanation of the Transfermarkt valuations. Page 5:

“Numerous studies have acknowledged Transfermarkt as a useful approximation to players’ market value. However, such a crowdsourcing approach has its limitations. In this respect, M ¨uller et al. [7] point out that the objectivity of the community members, on whose opinions the valuation is based, cannot be guaranteed. The process of aggregating user valuations lacks homogeneity, making it difficult to reproduce the results. Moreover, the estimation of market value tends to have higher variability for less popular players receiving less attention and valuations. More recently, Franceschi et al. [5] stress the subjectivity already mentioned by [7] and emphasize the conceptual difference between the transfer fee and the estimated market value provided by platforms like Transfermarkt. Nevertheless, researchers have recognized the empirical proximity between these measures [5, 7, 13, 23, 26, 27].” 

3) In their paper on the impact of Club revenues on transfer fees and salaries, Quansah et al. (2021) demonstrate that club revenues are a crucial variable in determining transfer fees and player market values, revealing the vulnerability of these factors during times of crises. Maybe it would be good to consider the implications of their findings for the relevance of utilizing Google Trends as a measure of popularity during periods of distress in your own research. 

Following this comment, we have included this concern as future research in the conclusions:

“To provide a complete view of the impact of players' popularity on their market value, future research should also include longitudinal data to capture both economic growth and crisis periods, as club revenues could significantly influence market values [54].” – Page 28

4) Furthermore, it is mentioned that Cristiano Ronaldo is the most followed person on Instagram. Lionel Messi, another popular but « aging » football figure, closely trails him in terms of followers. While there certainly had been a strong correlation between popularity and market value earlier in their careers, it appears that this relationship has become nonlinear. Although their popularity continues to grow, their market values have experienced significant declines. Additionally, Messi's transfer from PSG to Inter Miami in the US, a relatively inferior league, has garnered substantial media attention and had a noteworthy impact on the player's popularity (and google trends), which may not be (or is certainly not) reflected in the player's continuously decreasing market value. While these are only two examples of anecdotal evidence, there are certainly more cases.

This comment is really interesting. We agree that there is no linear relationship between popularity and market value. In fact, we believe that this is why the results of our work suggest that it is best studied with non-linear models.

5) The paper includes several unsupported claims, such as "However, Google Trends has been underused and misused by sports analysts." It is essential to provide evidence or support for such claims rather than making assertions without a factual basis. 

Thank you for your comment. To address this concern, we have carefully revised the wording of the paper to focus on the underutilization of Google Trends in the context of sports analysis, rather than making assertions about misuse. Furthermore, we would like to emphasize that our proposed approach offers improved methods for utilizing Google Trends more effectively in the estimation of players' valuation. 

Consequently, the referred sentence has been rewritten as: “However, Google Trends has been underused by sports analysts.”

Similarly, the statement "The sports industry is a place where most interests, from political interests to public interests, converge, and traditional media and social media have helped to increase football's popularity worldwide" remains unsupported, along with certain other claims in the document. There are instances where the reader fails to discern the source of the information presented.

Thank you for bringing this to our attention. We included the references used to support this affirmation in the revised version of the manuscript.

“The sports industry is a place where most interests, from political interests to public interests, converge, and traditional media and social media have helped to increase football's popularity worldwide [1,2].” – Page 2

[1] Galily, Y, Tamir, I, Limor, Y. Sports: Faster, Higher, Stronger, and Public

Relations. Hum. Aff. 2015 Mar;25(1):93–109.

[2] Seippel, Ø, Dalen, H B, Sandvik, M R, Solstad, G. M. From political sports to

sports politics: on political mobilization of sports issues. Int. J. Sport Policy

Politics. 2018 Mar;10(4):669–686.

6) In the introduction, it is stated that the "Big Five" European football leagues generated revenues exceeding €9.283 million in the 2018/2019 season. As we are currently in 2023 and the 2022/23 season has concluded, it is necessary to update this information.

Thank you for your comment; we have added to the text (page 2) the revenue generated in the 2022/2023 season. Also, please note that for the sake of consistency, we have kept the dates of the 2018/2019 season.

“According to the consulting company Deloitte, the 20 clubs with the highest turnover (all of which belong to the ‘Big Five' European football leagues) exceeded €9.200 million in revenue in the 2021/22 season, marginally below pre-COVID revenues of €9.283 million in 2018/19 [3].” – Page 2

7) Additionally, the data used in the analysis pertains to the 2018/19 season. It is important to clarify the reason for analyzing such outdated data. Was it due to convenience, as the initial research was conducted four years ago? 

We appreciate the reviewer's comment on the appropriateness of the data. We would like to clarify that the database was taken in the summer of 2019 because these data were used for a master's thesis. Note that this work is framed in the context of different analyses that comprise a PhD thesis. Therefore, for reasons of research and consistency, all analyses (the study of leagues, matches, and players) have been carried out with data from the 2018-2019 season. In addition, we would like to express that, despite the undoubted interest in having up-to-date data, this work aims to develop and provide a new methodology for calculating several popularity indicators. In this way, although the study has been done using football players to the season 2018-2019, this paper provides practical guidance for developing and incorporating the proposed indicators, which could be applied in sports analytics and any study in which popularity is relevant. Therefore, we hope that this new methodology is tested by other researchers and by ourselves by means of a wide range of data.

8) The statement "Thus, to solve the problem of the lack of information on the actual transfer fees, researchers and fans have started paying attention to websites that offer estimates of the market value of players" suggests that football fans have turned to websites like transfermarkt.de due to the scarcity of transfer fee information. It would be beneficial to provide evidence or support for this claim. 

Thank you again for pointing this out. Our references just support the use of these websites by researchers, so the mention to fans’ behavior has been removed. Citations to other research works using these websites have been included.

9) The information regarding Cristiano Ronaldo's 308 million followers on Instagram may also be outdated? The last time I looked he had close to 600 million followers.

Thanks for your suggestion. We have updated the information. Page 3

"A clear example is Cristiano Ronaldo, who in 2021 had 308 million followers on Instagram (making him the person with the most followers in the world), charged $1.6 million per sponsored post, and earns more than 40 million dollars per year from publications alone [14]. Now, two years later, his followers exceed 590 million and he is still the person with the most followers in the world."

---

## [Decision Letter · Decision Letter 2]

14 Jul 2023

Measuring the popularity of football players with Google Trends

PONE-D-22-31349R2

Dear Dr. Malagón-Selma,

We’re pleased to inform you that your manuscript has been judged scientifically suitable for publication and will be formally accepted for publication once it meets all outstanding technical requirements.

Kind regards,

Prof. Dr. Florian Follert

Academic Editor

PLOS ONE

Additional Editor Comments:

Congratulations!

Reviewers' comments:

Reviewer's Responses to Questions

**Comments to the Author**

1. If the authors have adequately addressed your comments raised in a previous round of review and you feel that this manuscript is now acceptable for publication, you may indicate that here to bypass the “Comments to the Author” section, enter your conflict of interest statement in the “Confidential to Editor” section, and submit your "Accept" recommendation.

Reviewer #2: All comments have been addressed

2. Is the manuscript technically sound, and do the data support the conclusions?

Reviewer #2: Yes

3. Has the statistical analysis been performed appropriately and rigorously? 

Reviewer #2: Yes

4. Have the authors made all data underlying the findings in their manuscript fully available?

Reviewer #2: No

5. Is the manuscript presented in an intelligible fashion and written in standard English?

Reviewer #2: Yes

6. Review Comments to the Author

Reviewer #2: The authors have done a fine job with their revision opportunity. The issues/concerns I had with the original submission and the re-submission have been effectively addressed. The paper has meaningfully improved, the points that lacked in the original and the re-submission have been solved.

Taken together then, the contribution the article offers to the literature is now overt and meaningful. Job well done!

7. PLOS authors have the option to publish the peer review history of their article (what does this mean?). If published, this will include your full peer review and any attached files.

Reviewer #2: No

---

## [Editor Report · Acceptance letter]

26 Jul 2023

PONE-D-22-31349R2 

Measuring the popularity of football players with Google Trends 

Dear Dr. Malagón-Selma:

I'm pleased to inform you that your manuscript has been deemed suitable for publication in PLOS ONE. Congratulations! Your manuscript is now with our production department. 

Kind regards, 

on behalf of

Prof. Dr. Florian Follert 

Academic Editor

PLOS ONE